# COMPOL: A Unified Neural Operator Framework for Scalable Multi-Physics Simulations

## Abstract

*Multi-physics simulations* play an essential role in accurately modeling complex interactions across diverse scientific and engineering domains. Although neural operators, especially the Fourier Neural Operator (FNO), have significantly improved computational efficiency, they often fail to effectively capture intricate correlations inherent in coupled physical processes. To address this limitation, we introduce COMPOL, a novel coupled multi-physics operator learning framework. COMPOL extends conventional operator architectures by incorporating sophisticated recurrent and attention-based aggregation mechanisms, effectively modeling interdependencies among interacting physical processes within latent feature spaces. Our approach is architecture-agnostic and seamlessly integrates into various neural operator frameworks that involve latent space transformations. Extensive experiments on diverse benchmarks—including biological reaction-diffusion systems, pattern-forming chemical reactions, multiphase geological flows, and thermo-hydro-mechanical processes — demonstrate that COMPOL consistently achieves superior predictive accuracy compared to state-of-the-art methods.

## 1 Introduction

Physical simulations governed by partial differential equations (PDEs) are fundamental tools across numerous scientific and engineering fields, including aerospace engineering, fluid mechanics, chemical processes, and environmental modeling (Reed & Simon, 1980; Arnol'd, 2013; Evans, 2022; Chorin et al., 1990; Bergman, 2011). These simulations leverage core physical principles, such as conservation laws and symmetries, to accurately predict complex phenomena. Traditional numerical methods, like finite difference, finite element, and finite volume techniques, have been extensively used but face significant computational challenges when addressing high-dimensional or coupled multi-physics problems (Quarteroni et al., 2010; Susanne et al., 1994; LeVeque, 2007; Hughes, 2003). Recent advances in machine learning have introduced neural operator methods (Kovachki et al., 2023; Lu et al., 2021; Li et al., 2020b;c), notably the Fourier Neural Operator

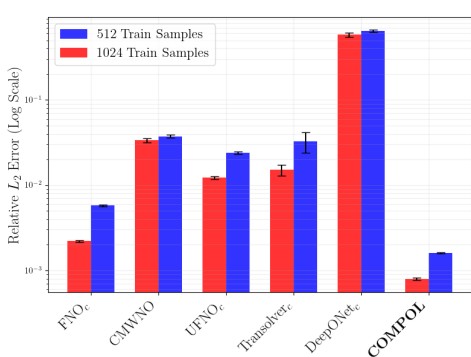

Figure 1: Predictive performance comparison of state-of-the-art neural operators vs. COMPOL on the 3-process Belousov–Zhabotinsky equations (512 and 1024 samples).

(FNO) (Li et al., 2020a), which directly approximate solution mappings between infinite-dimensional function spaces, significantly enhancing computational efficiency. Despite their promise, existing neural operator methods often inadequately capture the intricate correlations and dependencies present in coupled multi-physics systems.

Modeling multi-physics systems presents unique challenges due to the dynamic and complex interactions between multiple distinct physical processes, each governed by its own set of PDEs. These interactions often span multiple spatial and temporal scales, creating significant computational complexity and modeling difficulties (Keyes et al., 2013; Quarteroni & Quarteroni, 2009). Examples include fluid-structure interactions, chemical reaction-diffusion systems, and multiphase geological flows (Bazilevs et al., 2013; Bungartz & Schäfer, 2006; Weinan & Engquist, 2003; Cross & Green-

side, 2009; Holmes, 2012), *etc.* Current neural operator frameworks typically treat these processes independently or apply overly simplistic coupling strategies, resulting in insufficient representation of inter-process dynamics. Consequently, capturing the nuanced interactions and accurately predicting system behaviors remains a critical and unresolved challenge in neural operator-based multi-physics modeling.

To address these challenges, we propose COMPOL, a novel coupled multi-physics operator learning framework explicitly designed to effectively represent complex interactions among multiple physical processes. COMPOL introduces sophisticated recurrent (Cho et al., 2014) and attention (Vaswani, 2017) mechanisms for latent feature aggregation, capturing detailed inter-process interactions within latent representation spaces. Notably, our proposed framework is architecture-agnostic, capable of integration with various neural operator frameworks that involve latent feature transformations. Our contributions are as follows:

- Proposing a versatile, architecture-agnostic operator learning framework tailored for coupled multi-physics systems, enhancing traditional operator layers with sophisticated feature aggregation methods.
- Introducing flexible latent aggregation techniques, including recurrent and attention mechanisms, to robustly capture dynamic inter-process dependencies.
- Demonstrating COMPOL's significant effectiveness and scalability across diverse multi-physics benchmarks, including predator-prey dynamics, chemical reactions, multiphase flows, and thermo-hydro-mechanical processes. Unlike previous methods restricted to coupling only two processes, COMPOL scales flexibly to an arbitrary number of interacting physical processes, consistently achieving substantial improvements in predictive accuracy compared to state-of-the-art approaches.

Collectively, COMPOL establishes a robust, flexible, and generalizable approach, significantly advancing the modeling capabilities of neural operators for complex multi-physics simulations.

## 2 Background

**Operator Learning**  Operator learning (Kovachki et al., 2023) directly approximates solution operators of partial differential equations (PDEs) by learning mappings between infinite-dimensional function spaces. Consider a PDE defined as: $\mathfrak{L}(u)(x) = f(x), x \in \Omega; u(x) = 0, x \in \partial\Omega$ where $\mathfrak{L}$ is a differential operator, $u$ is the solution function, $f$ is the input function, $\Omega$ is the problem domain and $\partial\Omega$ represents its boundary. Operator learning aims to approximate the inverse operator $\mathfrak{L}^{-1}$ mapping inputs $f$ to $u$ through a parametric operator $\psi_\theta : \mathcal{H} \to \mathcal{Y}$. This is achieved by minimizing the empirical risk over a dataset consisting of function pairs $\{(f_n, u_n)\}_{n=1}^N$: $\theta^* = \arg\min_\theta \frac{1}{N} \sum_{n=1}^N \|\psi_\theta(f_n) - u_n\|_{\mathcal{Y}}^2$ where $\mathcal{H}$ and $\mathcal{Y}$ represent suitable function spaces, such as Banach or Hilbert spaces.

**Fourier Neural Operator (FNO)**  The Fourier Neural Operator (Li et al., 2020a) approximates the solution operator using spectral convolutions in the Fourier domain. Given an input function $v(x)$, FNO applies Fourier layers defined by: $v(x) \leftarrow \sigma\left(Wv(x) + \mathcal{F}^{-1}(R(k) \cdot \mathcal{F}(v(x)))\right)$, where $\mathcal{F}$ and $\mathcal{F}^{-1}$ denote the Fourier transform and inverse Fourier transform, respectively. The spectral kernel $R(k)$ is parameterized as a complex-valued tensor in Fourier space, which is learned during training. Specifically, given $v(x) \in \mathbb{R}^d$, the Fourier transform and its inverse transform are defined as $\mathcal{F}(v(x))(k) = \int_{\mathbb{R}^d} v(x)e^{-2\pi i k \cdot x}dx, \mathcal{F}^{-1}(\hat{v}(k))(x) = \int_{\mathbb{R}^d} \hat{v}(k)e^{2\pi i k \cdot x}dk.$ This spectral approach efficiently captures global correlations within the input function.

## 3 Method

### 3.1 Multi-Physics Simulation

We consider multi-physics systems governed by *coupled* partial differential equations (PDEs), which naturally emerge in diverse scientific and engineering disciplines, such as chemical reactions, environmental modeling, biological processes, and fluid mechanics. These systems involve multiple interacting physical processes, each described by its own PDE and interconnected through nonlinear coupling terms (Weinan & Engquist, 2003). Such interactions substantially influence the overall system behavior, creating considerable modeling and computational challenges. Formally, a general mathematical representation for a coupled PDE system involving $M$ interacting physical processes is

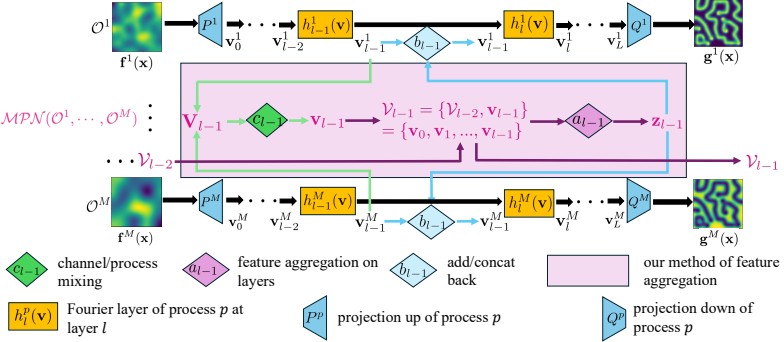

Figure 2: A graphical representation of COMPOL

given by:

$$\mathfrak{L}_m (u_1, u_2, \ldots, u_M) (x, t) = f_m(x, t), \quad m = 1, \ldots, M,$$
$$u_m(x, t) = 0, \quad x \in \partial\Omega_m, \tag{1}$$

where $\mathfrak{L}_m$ represents differential operators, potentially including spatial and temporal derivatives and nonlinear terms capturing interactions among the physical processes. Here, $u_m(x, t)$ are the unknown state variables associated with the $m$-th physical process, and $f_m(x, t)$ represent known source terms, external inputs, or boundary conditions for each process.

### 3.2 Coupled Multi-Physics Operator Learning (COMPOL)

Neural operator learning has shown great promise for modeling physical phenomena governed by PDEs. However, a significant limitation remains: most existing approaches assume that systems are governed by a single set of PDEs. This simplification overlooks the reality of physical simulations, where multiple processes interact across different scales, resulting in complex behaviors (Keyes et al., 2013). Existing techniques like feature concatenation (Wen et al., 2022) and cross-overlaying (Xiao et al., 2023) attempt to model these interactions but are often insufficient for capturing sophisticated multi-physics scenarios.

To overcome this limitation, we propose COMPOL, a coupled multi-physics neural operator, $\mathcal{MPO}(\mathcal{O}_1, \cdots, \mathcal{O}_M)$ designed specifically to capture the intricate correlations among individual operators $\mathcal{O}_1, \cdots, \mathcal{O}_M$ of a multi-physics system. COMPOL achieves this by leveraging advanced neural operator architectures combined with sophisticated latent feature aggregation mechanisms. Formally, COMPOL approximates solution operators for coupled PDE systems as mappings between function spaces: $\mathcal{G}_\theta : \mathcal{H}_1 \times \mathcal{H}_2 \times \cdots \times \mathcal{H}_M \to \mathcal{Y}_1 \times \mathcal{Y}_2 \times \cdots \times \mathcal{Y}_M$ where each $\mathcal{H}_m$ denotes the function space of input functions (such as initial or boundary conditions), and $\mathcal{Y}_m$ denotes the corresponding output solution function space for the $m$-th physical process. Given input functions $f^m(x)$ representing the initial conditions or external inputs for each physical process, we first embed each process into a shared latent space to facilitate consistent interaction modeling: $v_0^m(x) = P^m(f^m(x))$ where $P^m$ are linear projection operators mapping input functions into a common latent feature space, which enables unified treatment of heterogeneous processes.

Subsequently, latent features undergo iterative transformations using nonlinear multi-physics neural operator layers explicitly designed to capture complex cross-process interaction: $v_l^m(x) = h_l^m(v_{l-1}^m(x), z_{l-1}(x))$, $l = 1, \ldots, L$, where each neural operator layer $h_l^m$ comprises a nonlinear transformation that represents the internal process dynamics and is augmented by interaction-aware aggregated features $z_{l-1}(x)$. Specifically, the latent representation $v_l^m(x)$ captures the detailed inner-process transformations, encapsulating the *process-specific* evolution and *local dynamics* independently for each physical process. In contrast, the aggregated latent feature $z_{l-1}(x)$ explicitly represents *inter-process* interactions, integrating *global context* and information across different physical processes. This separation of inner-process transformations and inter-process interactions is a crucial design choice. It allows the model to independently refine representations of each process while simultaneously and explicitly modeling the intricate and nonlinear interactions among them.

The aggregated interaction features are computed using a carefully designed aggregation function: $z_{l-1}(x) = \mathcal{A}(v_{l-1}^1(x), v_{l-1}^2(x), \ldots, v_{l-1}^M(x))$, with $\mathcal{A}(\cdot)$ denoting a specialized aggregation mechanism (such as recurrent or attention), ensuring coherent and effective representation of inter-process dependencies. Finally, each process-specific solution is reconstructed from the latent features using dedicated linear projection operators: $g^m(x) = Q^m(v_L^m(x))$ yielding the predicted solutions $g^m(x)$ in their respective physical spaces.

**Universal Approximation Capabilities**   COMPOL is theoretically grounded in the operator approximation theory, particularly relying on the universal approximation property of neural operator frameworks. According to this theory, neural operators can uniformly approximate any continuous nonlinear operator between Banach spaces. Specifically, given a continuous operator $\mathcal{G} : \mathcal{K} \subset \mathcal{H} \to \mathcal{Y}$ defined on a compact set $\mathcal{K}$, and for any $\varepsilon > 0$, there exist neural operator parameters $\theta$ such that:

$$\sup_{f \in \mathcal{K}} \|\mathcal{G}(f) - \mathcal{G}_\theta(f)\|_y < \varepsilon.$$

**Convergence of the Iterative Latent Transformation**   In the context of COMPOL, the iterative transformations of latent representations can be interpreted as fixed-point iterative schemes: $v_l^m = \Phi_l^m \left( v_{l-1}^m, z_{l-1} \right), z_{l-1} = \mathcal{A} \left( \left\{ v_{l-1}^m \right\}_{m=1}^M \right),$

where $\Phi_l^m$ are nonlinear integral operators that encapsulate the process-specific inner transformations. Under appropriate conditions, such as Lipschitz continuity of $\Phi_l^m$ and $\mathcal{A}$, Banach's fixed-point theorem ensures convergence to a unique fixed point, enhancing stability and reliability of the neural operator approximation.

**Stability of Latent Feature Aggregation**   Moreover, the explicit distinction between inner-process transformations ($v_l^m$) and inter-process interactions ($z_l$) in COMPOL theoretically enhances expressivity and interpretability. The carefully designed aggregation mechanisms, such as gating in recurrent neural networks and adaptive weighting in attention mechanisms, provide structured inductive biases, facilitating robust and adaptive feature integration. These mechanisms inherently regularize the learning process and promote generalization by prioritizing and selectively integrating the most impactful cross-process interaction features. Thus, the explicit structural design of COMPOL offers a rigorous and theoretically justified approach for modeling complex multi-physics interactions.

Collectively, these theoretical results provide robust mathematical foundations for the COMPOL framework. The universal approximation theorem establishes that COMPOL can precisely model any complex multi-physics operator with arbitrary accuracy. The convergence theorem offers strong mathematical guarantees that COMPOL's iterative latent updates consistently converge, ensuring reliable and stable model predictions. Additionally, the stability theorem confirms that COMPOL robustly represents and integrates interactions between different physical processes, enhancing the interpretability and reliability of the learned latent representations. These rigorous theoretical underpinnings validate COMPOL's effectiveness, stability, and generalization capabilities when addressing complex, coupled multi-physics problems. We provide proofs in the Appendix A.

### 3.3 Aggregation of Multi-Physics Operator Layers

We now restate the formulation explicitly using discretized function notation. We consider a coupled physical system with $M$ processes, each associated with discretized input functions $\mathbf{f}^1, \ldots, \mathbf{f}^M$ (Figure 2). Each process $m$ first maps its input $\mathbf{f}^m$ to a latent representation $\mathbf{v}_0^m$ via a channel-wise linear layer $P^m : \mathbb{R}^{d_m} \to \mathbb{R}^{d_h}$, where $d_m$ and $d_h$ are input and latent dimensions, respectively. By stacking $L$ neural operator layers, each subsequent operator layer $h_l^m$ transforms latent representations $\mathbf{v}_{l-1}^m$ into $\mathbf{v}_l^m$. A final linear layer $Q^m : \mathbb{R}^{d_h} \to \mathbb{R}^{d_m}$ maps the latent representation $\mathbf{v}_L^m$ back to the output function $\mathbf{g}^m$. To capture inter-process interactions, we compute an aggregated latent state $\mathbf{z}_l = \mathcal{A}(\{\{\mathbf{v}_j^m\}_{m=1}^M\}_{j=0}^l)$ at each layer where aggregation mechanisms are implemented using either recurrent units (Cho et al., 2014) or attention-based (Vaswani, 2017) methods.

**Recurrent Aggregation State**   For notational convenience, let $\mathcal{V}$ represent the complete collection of intermediate latent features across $M$ processes and $L + 1$ layers, where these features are produced by neural operator layers $\{\{h_l^m\}_{m=1}^M\}_{l=0}^L$. Specifically, $\mathcal{V} = \{\{\mathbf{v}_l^m\}_{m=1}^M\}_{l=0}^L$. In the one-dimensional case, $\mathcal{V}$ can be viewed as a $(L + 1) \times M \times d_h$ tensor. From this structure, we define $\mathbf{V}_l$ as the $M \times d_h$ matrix obtained by extracting the $l$-th slice along the first dimension of $\mathcal{V}$. We further introduce $\mathcal{V}_l$ to represent the sequence of latent functions from layer 0 through layer $l$, which can be written as $\mathcal{V}_l = \{\mathbf{V}_j\}_{j=0}^l$. Within this notation, two cases are particularly noteworthy: $\mathbf{V}_0$ represents the collection of latent functions immediately following the channel-wise lifting operation, while $\mathbf{V}_L$ denotes the collection of all latent functions just before the channel-wise projection.

Our first attempt of computing the aggregation state of layer $l$ is with using Gated Recurrent Unit (GRU) (Cho et al., 2014), expressed as: $\mathbf{q}_l = \sigma(\mathbf{W}_q \mathbf{V}_l + \mathbf{U}_q \mathbf{z}_{l-1} + \mathbf{b}_q), \mathbf{r}_l = \sigma(\mathbf{W}_r \mathbf{V}_l + \mathbf{U}_r \mathbf{z}_{l-1} + \mathbf{b}_r), \tilde{\mathbf{z}}_l = \tanh(\mathbf{W}_z \mathbf{V}_l + \mathbf{U}_z(\mathbf{r}_l \odot \mathbf{z}_{l-1}) + \mathbf{b}_z), \mathbf{z}_l = \mathbf{q}_l \odot \mathbf{z}_{l-1} + (1 - \mathbf{q}_l) \odot \tilde{\mathbf{z}}_l$ where, $\mathbf{z}_{l-1}$ represents the aggregation state from the previous layer, , $\mathbf{V}_l$ captures the collective latent transformations across

all $M$ processes at layer $l$ and $\odot$ denotes element-wise multiplication. Computing the aggregation state $\mathbf{z}_l$ involves two intermediate states: the update gate $\mathbf{q}_l$, balancing historical and new information, and the reset gate $\mathbf{r}_l$, selectively filtering irrelevant past information. A candidate state $\tilde{\mathbf{z}}_l$ combines current inputs $\mathbf{V}_l$ with the filtered previous state $\mathbf{r}_l \odot \mathbf{z}_{l-1}$ (Jozefowicz et al., 2015; Pascanu, 2013). The final aggregation state $\mathbf{z}_l$ is a weighted blend of $\mathbf{z}_{l-1}$ and $\tilde{\mathbf{z}}_l$ controlled by the update gate $\mathbf{q}_l$. This approach effectively captures intricate interactions across layers, enabling the model to leverage complex dynamics in the coupled physical system.

**Attention Aggregation State**  In the attention-based aggregation mechanism, we compute the global latent interaction state through a scaled dot-product attention tailored specifically for multi-physics systems. Initially, the latent features from each process are linearly projected into query ($Q$), key ($K$), and value ($A$) representations via process-specific transformations: $Q = \Phi_Q(\mathcal{V}_l)$, $K = \Phi_K(\mathcal{V}_l)$, and $A = \Phi_A(\mathcal{V}_l)$. The aggregation state $\mathbf{z}_l$ is then derived as a weighted sum of these value vectors, where each weight $\alpha_j$ quantifies the significance of interactions involving the $j$-th process: $z_l = \sum_{j=1}^{M} \alpha_j A_j, \alpha_j = \frac{\exp\left(QK_j^\top / \sqrt{d_k}\right)}{\sum_{i=1}^{M} \exp\left(QK_i^\top / \sqrt{d_k}\right)}$ with $d_k$ representing the dimensionality of the key vectors. Thus, this approach explicitly encodes and integrates process-specific interactions, dynamically identifying and highlighting the most influential cross-process features at each layer.

**Complexity Analysis**  The computational complexity of the COMPOL model includes three components. Projection layers have complexity $O(Md_m d_h)$, with $M$ processes, input dimension $d_m$, and latent dimension $d_h$. Each Fourier Neural Operator (FNO) layer involves FFT and inverse FFT operations, each with complexity $O(N \log N)$ per channel (Cooley & Tukey, 1965), where $N$ denotes spatial discretization points, leading to a total complexity of $O(LMd_h \cdot N \log N)$ for $L$ layers. The aggregation mechanisms contribute additional complexity: $O(LM^2 d_h)$ for attention-based aggregation and $O(LMd_h^2)$ for GRU-based aggregation. Overall, the complexity is dominated by the Fourier operations, $O(LMd_h \cdot N \log N)$, in typical scenarios where $N \log N \gg M, d_h$. Aggregation mechanisms significantly influence complexity only when $M$ or $d_h$ is exceptionally large.

**Generalizability to Other Neural Operator Frameworks**  COMPOL builds upon the Fourier Neural Operator (FNO) due to its proven efficiency and strong performance across diverse tasks. Nonetheless, the proposed latent aggregation mechanisms are general and can be seamlessly integrated with other neural operator frameworks, such as DeepONet (Lu et al., 2021), GNO (Li et al., 2020b), LNO (Kovachki et al., 2023), and transformer-based operators (Wu et al., 2024) (e.g., Transolver). We provide further discussion and illustrative implementations for extending COMPOL to these alternative frameworks in the Section B.

### 3.4 Training

Given the training data that are simulated or sampled from a coupled multi-physics system, denoted as $\{\{\mathbf{f}_n^m, \mathbf{y}_n^m\}_{n=1}^{N_m}\}_{m=1}^M$, where $\mathbf{f}_n^m$ and $\mathbf{y}_n^m$ represent the $n$-th input and output functions, respectively, from the $m$-th process of the coupled system, we optimize our coupled multi-physics neural operator ($\mathcal{MPO}$) by minimizing the empirical risk, $\mathcal{L}_{\mathcal{MPO}} = \mathbb{E}_{m \sim \pi} \mathbb{E}_{f^m \sim \mu^m} ||\mathcal{MPO}(f) - y^m|| \approx \frac{1}{M} \sum_{m=1}^{M} \frac{1}{N_m} \sum_{n=1}^{N_m} ||\mathbf{g}_n^m - \mathbf{y}_n^m||$ where $\mathbf{g}_n^m$ is the prediction of $\mathbf{f}_n^m$. We can then use any gradient-based optimization method to minimize $\mathcal{L}_{\mathcal{MPO}}$.

## 4 Related Work

Operator learning represents an innovative approach to surrogate modeling that maps input functions to output functions. Traditional surrogate modeling methods have typically focused on mapping limited sets of system parameters *e.g.,* PDE parameters to output functions *e.g.,* PDE solution functions, as demonstrated in numerous research works (Higdon et al., 2008; Zhe et al., 2019; Li et al., 2021; Wang et al., 2021; Xing et al., 2021b;a; Li et al., 2022a). Neural operator methods, which leverage neural networks as their foundational architecture, have driven substantial progress in operator learning. The Fourier Neural Operator (FNO) (Li et al., 2020a) emerged alongside a simpler variant, the Low-rank Neural Operator (LNO) (Kovachki et al., 2023), which utilizes a low-rank decomposition of the operator's kernel to enhance computational efficiency. Building on these innovations, researchers introduced the Graph Neural Operator (GNO) (Li et al., 2020b), which innovatively combines Nystrom approximation with graph neural networks to approximate function convolution. The Multipole Graph Neural Operator (MGNO) (Li et al., 2020c) introduced a multi-scale kernel decomposition approach, achieving linear computational complexity in convolution calculations. Parallel developments included a multiwavelet-based operator learning model (Gupta

et al., 2021), which enhanced precision through fine-grained wavelet representations of the operator's kernel. Another significant contribution emerged with the Deep Operator Net (DeepONet) (Lu et al., 2021), which employs a dual-network architecture combining a branch network for input functions and a trunk network for sampling locations. This architecture was later refined into the POD-DeepONet (Lu et al., 2022), which improved stability and efficiency by replacing the trunk network with POD (or PCA) bases derived from training data. A survey of neural operators is given in (Kovachki et al., 2023). Recent advances in operator learning have focused on developing mesh-agnostic and data-assimilation approaches, representing a significant departure from traditional methods (Yin et al., 2022; Chen et al., 2022; Pilva & Zareei, 2022; Boussif et al., 2022; Esmaeilzadeh et al., 2020). These novel approaches remove the constraint of requiring input and output functions to be sampled on fixed or regular meshes, offering greater flexibility in handling diverse data structures. A key innovation in these methods lies in their reformulation of PDE simulation as an ODE-solving problem. This fundamental shift changes the primary objective from simply predicting solutions based on initial or boundary conditions to capturing the complete dynamic behavior of PDE systems. Training PDE surrogates and neural operators requires extensive high-quality data, which poses a significant challenge due to the resource-intensive nature of physical experiments and sophisticated simulators. To optimize predictive performance while minimizing data collection costs, researchers employ multi-fidelity, multi-resolution modeling Tang et al. (2024; 2023); Li et al. (2023), and active learning approaches (Li et al., 2022b; 2024).

Our approach aligns with recent multi-physics neural operator research, though it addresses distinct goals and employs different strategies (McCabe et al., 2023; Rahman et al., 2024; Hao et al., 2024). Unlike existing works focusing on general physics surrogate models using diverse simulation data, we specifically target interactions within individual coupled physical systems. Closely related is the Coupled Multiwavelet Neural Operator (CMWNO) (Xiao et al., 2023), which models coupled PDEs using wavelet-based methods. However, CMWNO has limitations, including rigid decomposition schemes, restricted coupling structures, and difficulty handling multiple interacting processes effectively. Our proposed method builds upon flexible FNO layers, enhancing adaptability and extensibility. The introduced aggregation mechanism integrates seamlessly with any latent-feature neural operator framework and naturally supports complex interactions across multiple physical processes.

| Method | Train Samples | Lotka-V. | Belousov-Z. | Grey-Scott | Multiphase | THM |
|--------|---------------|----------|-------------|------------|------------|-----|
| FNO$_c$ | 512 | 0.3251 ±0.0507 | 0.0058 ±1.4e-4 | 0.0056 ±0.0001 | 0.0232 ±0.0005 | 0.1826 ±0.0088 |
| | 1024 | 0.1499 ±0.0097 | 0.0022 ±4.4e-5 | 0.0042 ±7.8e-5 | 0.0684 ±0.1071 | 0.1577 ±0.0073 |
| CMWNO | 512 | 0.3989 ±0.0422 | 0.0375 ±0.0015 | - | - | - |
| | 1024 | 0.3876 ±0.0357 | 0.0336 ±0.0019 | - | - | - |
| UFNO$_c$ | 512 | 0.2519 ±0.0518 | 0.0239 ±7.7e-4 | 0.0083 ±0.0003 | 0.0241 ±0.0004 | 0.1824 ±0.0094 |
| | 1024 | 0.1677 ±0.0120 | 0.0122 ±5.1e-4 | 0.0044 ±9.8e-5 | 0.0137 ±0.0010 | 0.1496 ±0.0086 |
| Transolver$_c$ | 512 | 0.2606 ±0.0515 | 0.0329 ±0.0089 | 0.0251 ±0.0009 | 0.0503 ±0.0311 | 0.2176 ±0.0123 |
| | 1024 | 0.1262 ±0.0348 | 0.0151 ±0.0023 | 0.0174 ±0.0009 | 0.0185 ±0.0037 | 0.1916 ±0.0126 |
| DeepONet$_c$ | 512 | 0.4024 ±0.0143 | 0.6491 ±0.0199 | 0.7655 ±0.0039 | 0.1123 ±0.0051 | 0.2867 ±0.0199 |
| | 1024 | 0.3005 ±0.0285 | 0.5810 ±0.0352 | 0.6700 ±0.0240 | 0.0766 ±0.0049 | 0.2388 ±0.0125 |
| COMPOL-RNN | 512 | 0.0920 ±0.0085 | 0.0033 ±2.6e-4 | 0.0062 ±0.0039 | 0.0225 ±0.0108 | **0.1607 ±0.0089** |
| | 1024 | 0.0531 ±0.0100 | 0.0015 ±9.7e-5 | 0.0035 ±8.0e-5 | 0.0110 ±0.0010 | **0.1407 ±0.0097** |
| COMPOL-ATN | 512 | **0.0918 ±0.0042** | **0.0016 ±2.7e-5** | **0.0050 ±0.0016** | **0.0182 ±0.0011** | 0.1647 ±0.0092 |
| | 1024 | **0.0510 ±0.0048** | **0.0008 ±2.6e-5** | **0.0035 ±7.2e-5** | **0.0110 ±0.0009** | 0.1442 ±0.0066 |

Table 1: Comparison of relative $L_2$ prediction errors (mean ± standard deviation over 5-runs) for COMPOL variants (COMPOL-RNN, COMPOL-ATN) against baseline neural operators (FNO, UFNO, Transolver, DeepONet, CMWNO) across five multi-physics benchmark datasets, evaluated with 512 and 1024 training samples. Bold values indicate the best performance for each case.

## 5 Experiment

**Benchmarks** We evaluated our framework's performance on predicting solutions for coupled multi-physics PDE systems using five benchmark test cases: 1-D Lotka-Volterra equations (Murray, 2007) (2 processes), 1-D Belousov-Zhabotinsky equations (Petrov et al., 1993) (3 processes), 2-D Gray-Scott equations (Pearson, 1993) (2 processes), 2-D multiphase flow of oil and water in geological storage (Bear & Cheng, 2010; Hashemi et al., 2021; Abou-Kassem et al., 2013) (2 processes), and a 2-D Thermal-Hydro-Mechanical problem Gao & Ghassemi (2020)(5 processes considering

strain components separately). These diverse equations represent broad scientific and engineering applications. Detailed descriptions of these benchmarks are provided in the Appendix C, D and E Training datasets were generated with numerical solvers using 256-point meshes for 1-D cases and 64×64 meshes for 2-D cases. We trained models on two dataset sizes (512 and 1024 samples) and evaluated performance on an independent test set of 200 examples. Coefficients and boundary conditions remained constant to isolate effects of varying initial conditions. The primary objective was to assess our framework's capability to map initial conditions ($t = 0$) to final solution fields ($t = T$), highlighting its ability to model complex interactions and dynamics.

**Competing Methods** To evaluate our proposed coupled multi-physics neural operator framework, we compare two variants—COMPOL-RNN (recurrent aggregation) and COMPOL-ATN (attention-based aggregation)—against state-of-the-art neural operators: FNO (Li et al., 2020a), UFNO (Wen et al., 2022), Transolver (Wu et al., 2024), DeepONet Lu et al. (2021), and CMWNO (Xiao et al., 2023), each using their official implementations. Except for CMWNO, these baselines are not inherently designed for multi-physics modeling. Therefore, we adapt them by concatenating process inputs and outputs across their respective channels (e.g., $FNO_c$). For consistency, all models were implemented in PyTorch (Paszke et al., 2019) and trained with the Adam optimizer (Diederik, 2014) (learning rate = 0.001). Except for CMWNO, which employed a step scheduler (as in its original implementation), all models used a cosine annealing schedule (Loshchilov & Hutter, 2016). Models were trained for 500 epochs, sufficient for convergence in all cases except DeepONet, which required 10,000 epochs. Experiments were conducted on NVIDIA A100 GPUs. Robustness was ensured via 5-fold cross-validation, with performance evaluated using average relative $L_2$ error and corresponding standard deviations across folds. Due to implementation constraints, CMWNO experiments were limited to 1-D scenarios.

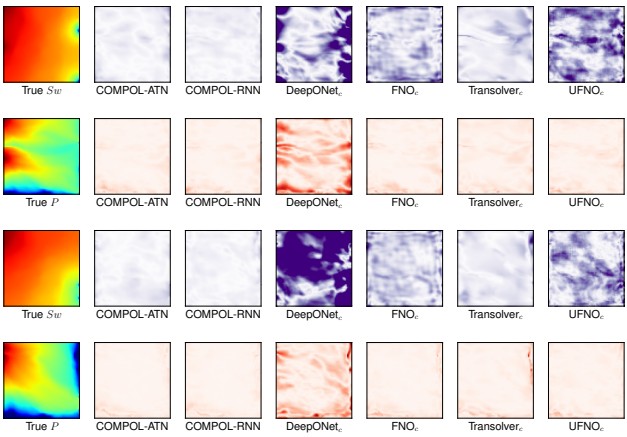

Figure 3: Element-wise solution error of coupled multiphase flow. The left most column is the ground-truth phase pressure $P$ and saturation $S_w$. The other columns are the absolute errors of each method. The lighter the color, the smaller the error.

## 5.1 Analysis of Predictive Performance

The experimental results in Table 1 present the predictive performance of COMPOL-RNN (recurrent aggregation) and COMPOL-ATN (attention-based aggregation) compared to baseline neural operators (FNO, UFNO, Transolver, DeepONet, and CMWNO), evaluated separately for 512 and 1024 training samples.For 512 training samples, COMPOL-ATN consistently achieves the lowest prediction errors, outperforming baseline methods with improvements up to 72.41%. COMPOL-RNN also demonstrates strong performance, showing improvements up to 63.48%, though slightly underperforming on the Grey-Scott dataset. When increasing the training set size to 1024 samples, COMPOL-ATN maintains superior performance, achieving improvements up to 63.64% compared to baselines, notably excelling in Lotka-Volterra and Belousov-Zhabotinsky datasets. COMPOL-RNN also continues robust performance with improvements up to 57.92% across benchmarks. Baseline neural operators adapted for multi-physics tasks exhibit significantly higher errors at both training scales. CMWNO, despite being explicitly designed for multi-physics modeling, consistently underperforms relative to both COMPOL variants.These results affirm the superior effectiveness and scalability of COMPOL methods in multi-physics modeling scenarios. We also provide a visualization of the element-wise

errors for the 2-D multiphase flow in Figure 3, along with a comparative visualization of predictions for the 1-D benchmarks in Section F.

## 5.2 Ablation Study of Coupling Mechanisms

To elucidate the factors underlying COMPOL's superior performance, we dissect its coupling mechanism into three core components: feature aggregation ($a_{l-1}$), augmentation ($b_{l-1}$), and mixing ($c_{l-1}$), as depicted in Figure 2. A detailed evaluation of each component's impact is provided in the Appendix G. Here, we address two crucial questions.

| Method | Modes | Width | Layers | Params Count (M) | BZ | LV |
|---|---|---|---|---|---|---|
| $FNO_c$ (default) | 16 | 64 | 4 | 0.5832 | $0.0022 \pm 4.4e\text{-}5$ | $0.1499 \pm 0.0097$ |
| $FNO_c$ | 96 | 64 | 4 | 3.2047 | $0.0018 \pm 4.9e\text{-}5$ | $0.1183 \pm 0.0106$ |
| $FNO_c$ | 64 | 64 | 6 | 3.2296 | $0.0020 \pm 4.4e\text{-}5$ | $0.1303 \pm 0.0152$ |
| $FNO_c$ | 48 | 64 | 8 | 3.2546 | $0.0032 \pm 0.0001$ | $0.1488 \pm 0.0160$ |
| $FNO_c$ | 16 | 224 | 2 | 3.6169 | $0.0014 \pm 2.6e\text{-}5$ | $0.1757 \pm 0.0102$ |
| $FNO_c$ | 16 | 128 | 6 | 3.4774 | $0.0018 \pm 3.8e\text{-}5$ | $0.1761 \pm 0.0099$ |
| $FNO_c$ | 16 | 160 | 4 | 3.6392 | $0.0016 \pm 4e\text{-}5$ | $0.1698 \pm 0.0090$ |
| COMPOL-RNN | 16 | 64 | 4 | BZ: 3.0749 LV: 2.4880 | $0.0015 \pm 9.7e\text{-}5$ | $0.0531 \pm 0.0100$ |
| COMPOL-ATN | 16 | 64 | 4 | BZ: 2.1029 LV: 1.4925 | $\mathbf{0.0008 \pm 2.6e\text{-}5}$ | $\mathbf{0.0510 \pm 0.0048}$ |

Table 2: Comparison of testing relative $L_2$ errors for FNO variants expanded to match COMPOL's parameter count versus COMPOL-RNN and COMPOL-ATN, evaluated on Belousov–Zhabotinsky (BZ) and Lotka–Volterra (LV) equations. Results show that COMPOL's superior performance is primarily due to its coupling mechanism rather than merely increased parameters.

**Capacity vs. Coupling: Parameter-Matched FNO**  COMPOL instantiates a separate neural operator per physical process and aggregates cross-process information in latent space; consequently, its parameter count scales roughly linearly with the number of processes $M$ relative to a single FNO with naïve channel concatenation. To test whether our gains are merely due to capacity, we scale FNO to match COMPOL ' parameter count by increasing Fourier modes, network width, and depth. On Belousov–Zhabotinsky (BZ; 3 processes) and Lotka–Volterra (LV; 2 processes) with 1024 samples, Table 2 reports relative $L_2$ errors. While the parameter-matched FNO improves over its base configuration, further widening or deepening yields diminishing returns and does not close the gap to COMPOL, indicating that the advantage primarily stems from explicit, scalable cross-process coupling rather than sheer size.

**Improvement over Single-Process Simulations**  We further examine whether COMPOL's latent aggregation mechanisms can enhance predictive performance for single-process simulations. Specifically, we evaluate COMPOL on the single-process Burgers' equation using training sets of 512 and 1024 examples, comparing results against baseline methods (Table 3). Results show that COMPOL significantly improves predictive accuracy, even in single-process scenarios, by effectively leveraging advanced latent aggregation techniques (attention-based and recurrent methods). These mechanisms enrich latent feature representations, implicitly regularize learning, adaptively select relevant features,

| Method | ntrain=512 | ntrain=1024 |
|---|---|---|
| FNO | $0.0715 \pm 0.0033$ | $0.0422 \pm 0.0037$ |
| CMWNO | $0.4441 \pm 0.0134$ | $0.4509 \pm 0.0168$ |
| UFNO | $0.1043 \pm 0.0061$ | $0.0565 \pm 0.0012$ |
| Transolver | $0.1264 \pm 0.0733$ | $0.4066 \pm 0.4248$ |
| DeepONet | $0.4007 \pm 0.0216$ | $0.3749 \pm 0.0181$ |
| COMPOL-RNN | $0.0397 \pm 0.0030$ | $0.0197 \pm 0.0013$ |
| COMPOL-ATN | $\mathbf{0.0200 \pm 0.0021}$ | $\mathbf{0.0106 \pm 0.0006}$ |

Table 3: Comparison of prediction accuracy (relative $L_2$ error) for the single-process Burgers' equation across methods trained with 512 and 1024 examples. Results averaged over 5 runs (500 epochs each), highlighting superior performance of COMPOL with latent aggregation.

and better capture underlying temporal dynamics. Consequently, COMPOL demonstrates improved optimization stability, superior generalization, and increased overall accuracy.

**Out-of-distribution generalization.**  We probe robustness under genuine distribution shift on the Belousov–Zhabotinsky (BZ) system while keeping the fixed-grid operator backbone (FNO). At test time, we perturb the initial-condition distribution by sampling Gaussian random fields (GRFs) with correlation length scales and spectral content not seen during training, plus additive noise. All models are trained on GRF length scale $l = 0.02$ and evaluated on $l \in \{0.015, 0.018\}$. Table 4 reports mean $\pm$ std across runs (lower is better). COMPOL maintains state-of-the-art accuracy under both shifts,

| Model | BZ (u, v, w) | | | GS (u, v) | |
|---|---|---|---|---|---|
| | u | v | w | u | v |
| CMWNO | 0.6517 | 0.7629 | 0.5010 | – | – |
| DeepONet | 2.0934 | 3.4638 | 2.7937 | 31.1403 | 30.5694 |
| FNO | 0.1475 | 0.0930 | 0.1182 | 0.4024 | 0.2256 |
| Transolver | 0.3899 | 0.4728 | 0.2775 | 0.4859 | 0.3413 |
| UFNO | 0.2210 | 0.1786 | 0.1072 | 0.4056 | 0.2734 |
| COMPOL-ATN | **0.1403** | **0.0672** | 0.1091 | 0.4037 | 0.2147 |
| COMPOL-RNN | 0.1479 | 0.0759 | **0.1052** | **0.3923** | **0.2082** |

Table 5: MAE of spectral energy distributions for BZ and GS (lower is better). Entries compare each model's spectrum to ground truth; best per column in **bold**. "–" denotes not reported.

| System | COMPOL (sec) | Numerical solver (sec) | Speedup |
|---|---|---|---|
| GS | 0.005 | 0.103 | 20.6× |
| BZ | 0.002 | 0.140 | 70× |
| Multiphase Flow | 0.0047 | 38.640 | 8221× |
| THM | 0.015 | 97.170 | 6478× |

Table 6: Per-sample inference time. **Speedup** is (solver time)/(COMPOL time); higher is better.

with COMPOL-ATN and COMPOL-RNN tying for the best errors at each OOD length scale, indicating strong generalization to changes in the spatial structure of the initial conditions.

**Predictive Spectrum Analysis.** Energy spectra offer a physics-informed diagnostic of scale-wise fidelity in the Fourier domain. For each state variable, we compute the mean absolute error (MAE) between predicted and ground-truth spectral energy distributions (lower is better). As summarized in Table 5, COMPOL matches the ground-truth spectra most closely across both Belousov–Zhabotinsky (BZ; $u, v, w$) and Gray–Scott (GS; $u, v$) systems. Relative to FNO, COMPOL-ATN reduces spectral MAE on BZ by $\sim$5% for $u$ and

| Model | $l = 0.015$ | $l = 0.018$ |
|---|---|---|
| FNO | $0.0940 \pm 0.0036$ | $0.0600 \pm 0.0021$ |
| COMPOL-RNN | $\mathbf{0.0935 \pm 0.0035}$ | $\mathbf{0.0597 \pm 0.0022}$ |
| COMPOL-ATN | $\mathbf{0.0935 \pm 0.0035}$ | $\mathbf{0.0597 \pm 0.0022}$ |
| UFNO | $0.0966 \pm 0.0033$ | $0.0639 \pm 0.0018$ |
| Transolver | $0.1256 \pm 0.0073$ | $0.0818 \pm 0.0060$ |
| DeepONet | $0.7279 \pm 0.0181$ | $0.7132 \pm 0.0218$ |
| CMWNO | $0.1065 \pm 0.0029$ | $0.0742 \pm 0.0018$ |

Table 4: OOD robustness on BZ under GRF length scale shifts. Mean $\pm$ std; lower is better; best per column in **bold**. Training $l = 0.02$.

$\sim$28% for $v$, while COMPOL-RNN yields the best $w$ with an $\sim$11% drop. On GS, COMPOL-RNN improves over FNO by $\sim$2.5% for $u$ and $\sim$7.7% for $v$, whereas other baselines lag substantially (e.g., Transolver and DeepONet). These consistent gains indicate that coupling mechanisms in COMPOL help match the ground-truth spectral content across variables and systems.

**Amortized inference cost and speedups.** We compare per-sample wall-clock inference of COMPOL against a finite-difference solver on the same 80-core CPU node. For fairness, all COMPOL runs are CPU-only (no GPU). Table 6 reports raw times and speedups, defined as solver time / COMPOL time (higher is better). Across Gray–Scott and Belousov–Zhabotinsky, COMPOL attains 20–70× acceleration, and reaches three to four orders of magnitude on the multiphysics cases, enabling large parameter sweeps and near–real-time evaluation.

# 6 Conclusion

We have introduced COMPOL, a novel coupled multi-physics neural operator learning framework that extends Fourier neural operators to effectively model interactions between multiple physical processes in complex systems. Our approach employs innovative feature aggregation techniques using recurrent and attention mechanisms to capture rich interdependencies in the latent space. Extensive experiments demonstrate that COMPOL achieves significant improvements in predictive accuracy compared to state-of-the-art methods. These results highlight the effectiveness of our feature aggregation approach for learning the complex dynamics of coupled multi-physics systems.

**Reproducibility Statement.** All details required to reproduce our results are included within the paper and the appendices. Specifically: model architectures, training hyperparameters (optimizer, learning rate schedule, number of epochs, batch sizes, FNO mode counts / widths / depths, settings for COMPOL-RNN and COMPOL-ATN) are given in §Method and §Training; the PDE systems used (equations, coupling terms, diffusion coefficients, initial/boundary conditions, spatial/temporal resolution) and data generation procedures for all benchmark tasks (Lotka–Volterra, Belousov–Zhabotinsky, Gray–Scott, Multiphase Flow, THM) appear in Appendices C, D, and E; the ablation experiments including parameter-matched FNO baselines are described in Table 2 and further detailed in Appendix G; out-of-distribution evaluation protocol (perturbed initial condition distributions via GRFs with unseen correlation lengths plus noise) is described alongside results in Table 4; spectral-energy error metrics are defined and evaluated in Table 5; timing measurements (inference time, speedups over numerical solvers) are set out in Table 6. The theoretical claims (universal approximation, convergence, stability) are fully stated and proven in Appendix A. Using only the information contained in these sections, an independent researcher should be able to reproduce the experiments, metrics, and tables/figures.

**The Use of Large Language Models (LLMs)** We used an LLM-based assistant solely for copy-editing and phrasing improvements. The model did not generate research ideas, design experiments, analyze data, or contribute substantive content. All technical contributions and conclusions are the authors' own; all edits were reviewed and verified by the authors. No confidential data beyond the manuscript text were provided, and this usage complies with the ICLR Code of Ethics.

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

# A   Proofs of Theoretical Claims for COMPOL

**Theorem A.1.** *(Universal Approximation of Neural Operators): Given a compact set $\mathcal{K} \subset \mathcal{H}$ and a continuous nonlinear operator $\mathcal{G} : \mathcal{K} \to \mathcal{Y}$, for every $\epsilon > 0$ there exists a parameter set $\theta$ of the neural operator $\mathcal{G}_\theta$, such that:*

$$\sup_{f \in \mathcal{K}} \|\mathcal{G}(f) - \mathcal{G}_\theta(f)\|_{\mathcal{Y}} < \varepsilon.$$

*Proof.* This theorem directly follows from the universal approximation theorem for operators stated and proven by Kovachki et al. (2023). Neural operator architectures, such as the Fourier Neural Operator and DeepONet, have been rigorously proven to possess universal approximation capabilities in appropriate Banach spaces.

Specifically, the proof leverages the density of neural network-generated basis functions in function spaces, using the Stone-Weierstrass theorem to approximate continuous operators with arbitrary precision. Given that COMPOL leverages neural operators with additional latent aggregation mechanisms, the universal approximation capacity remains unaffected, as these aggregation mechanisms are continuous mappings and thus preserve the universal approximation property. $\square$

**Theorem A.2.** *(Convergence of Iterative Latent Feature Transformation): Consider the iterative latent transformation defined by:*

$$v_l^m = \Phi_l^m \left( v_{l-1}^m, z_{l-1} \right), \quad z_{l-1} = \mathcal{A}\left( \left\{ v_{l-1}^m \right\}_{m=1}^M \right).$$

*If $\Phi_l^m$ and $\mathcal{A}$ are Lipschitz continuous with Lipschitz constants $L_\Phi$ and $L_\mathcal{A}$, respectively, and if $L_\Phi L_\mathcal{A} < 1$, then the iterative scheme converges to a unique fixed point.*

*Proof.* By the Banach fixed-point theorem, convergence to a unique fixed point is guaranteed if the combined iterative mapping is a contraction. Define the combined operator $\mathcal{T}$ such that:

$$\mathcal{T}(\mathbf{v}) := \Phi(\mathbf{v}, \mathcal{A}(\mathbf{v})), \quad \mathbf{v} := \{v^m\}_{m=1}^M.$$

For in the latent space, we have:

$$\|\mathcal{T}(\mathbf{v}) - \mathcal{T}(\mathbf{w})\| \leq L_\Phi \|\mathbf{v} - \mathbf{w}\| + L_\Phi L_\mathcal{A} \|\mathbf{v} - \mathbf{w}\| = L_\Phi \left(1 + L_\mathcal{A}\right) \|\mathbf{v} - \mathbf{w}\|.$$

If $L_\Phi \left(1 + L_\mathcal{A}\right) < 1$, $\mathcal{T}$ is is a contraction, and thus by the Banach fixed-point theorem, the iterative procedure converges to a unique fixed point in the latent representation. $\square$

**Theorem A.3.** *(Stability of Aggregation Mechanisms): The aggregation mechanism $\mathcal{A}(\cdot)$, either recurrent-based or attention-based, preserves stability in latent space representation provided that all internal parameters remain bounded and activations are Lipschitz continuous.*

*Proof.* Aggregation mechanisms such as RNNs and attention modules can be expressed as compositions of linear transformations and Lipschitz continuous nonlinearities (e.g., sigmoid, tanh, softmax). Since compositions of Lipschitz continuous functions with bounded parameters remain Lipschitz continuous and bounded, the aggregation mechanism:

$$z = \mathcal{A}\left( v^1, v^2, \ldots, v^M \right)$$

remains stable. Specifically, given Lipschitz continuity of all intermediate functions with constants $L_i$, we have:

$$\|\mathcal{A}(\mathbf{v}) - \mathcal{A}(\mathbf{w})\| \leq \left( \prod_i L_i \right) \|\mathbf{v} - \mathbf{w}\|,$$

indicating the stability of aggregation as long as all parameters and operations remain bounded and well-conditioned. $\square$

# B   Adapting COMPOL to Alternative Neural Operator Backbones

COMPOL enhances neural operators by explicitly modeling latent interactions among multiple physical processes. Given $M$ coupled processes, latent features at layer $l$ are updated as:

$$v_l^m(x) = h_l^m \left( v_{l-1}^m(x), z_{l-1}(x) \right), \quad m = 1, \ldots, M,$$

where the latent aggregation $z_{l-1}(x)$ is:

$$z_{l-1}(x) = \mathcal{A} \left( v_{l-1}^1(x), v_{l-1}^2(x), \ldots, v_{l-1}^M(x) \right).$$

## B.1   DeepONet + COMPOL

Original DeepONet (Lu et al., 2021) approximates an operator $G : f \mapsto g$:

$$G(f)(y) \approx \sum_{j=1}^{p} b_j(f) \cdot t_j(y),$$

where $b_j$ and $t_j$ represent branch and trunk neural networks. COMPOL introduces latent aggregation $z$ to enhance interactions between the branch and trunk networks, explicitly modeling their latent interdependencies:

- Aggregate branch-trunk latent features:

$$z = \mathcal{A} \left( b_j(f), t_j(y) \right)$$

- Modified DeepONet prediction

$$G(f)(y) \approx \sum_{j=1}^{p} b_j'(f, z) \cdot t_j'(y, z)$$

## B.2   GNO + COMPOL

Graph Kernel Network (GKN) has been adopted by existing works for partial differential equations Li et al. (2020b), while the evolution of latent features is typically local-each update depends only on the current layer's hidden state and neighborhood structure. Therefore, such a design overlooks the rich temporal evolution and accumulated semantics across layers. By contrast, COMPOL introduces a layer-wise aggregated latent representation $z_l$ that synthesizes information from all prior hidden states $\left\{ v_j^{(m)} \right\}_{j=0}^{l-1}$ across all processes $m$. Formally, the adapted message passing operator is given as

$$v_{l+1}^{(m)}(x) = \sigma \left( W v_t^{(m)}(x) + \frac{1}{|\mathcal{N}(x)|} \sum_{y \in \mathcal{N}(x)} \kappa_\phi(e(x,y)) v_l^{(m)}(y) + W_z z_t(x) \right),$$

where $W_z$ represents a learnable matrix enforcing a linear transformation to align dimensions of $z_l(x)$ with $W v_l^{(m)}(x)$. Such an adaptation for GKN allows the message passing at each spatial location to dynamically adjust its representation based on both the multi-process coupling and the full history of the model's latent dynamics.

## B.3   LNO + COMPOL

Low-rank Neural Operator (LNO) (Kovachki et al., 2023) approximates operators via low-rank decompositions:

$$v_l(x) = \sum_{k=1}^{K} \alpha_k(x) \beta_k(x)$$

where $\alpha_k, \beta_k$ represent latent low-rank features. COMPOL aggregation explicitly models inter-rank interactions:

- Aggregation step:

$$z_{l-1}(x) = \mathcal{A} \left( \alpha_k(x), \beta_k(x) \right).$$

- Modified LNO:

$$v_l(x) = \sum_{k=1}^{K} \alpha_k' \left( x, z_{l-1}(x) \right) \beta_k' \left( x, z_{l-1}(x) \right).$$

### B.4 Transformer-based Neural Operators (*e.g.,* Transolver) + COMPOL

Transformer-based neural operators perform positional self-attention:

$$v_l(x_i) = \text{Attention}(Q_i, K_j, V_j), \quad Q_i = Q(v_{l-1}(x_i)), K_j = K(v_{l-1}(x_j)), V_j = V(v_{l-1}(x_j))$$

To integrate COMPOL into transformer-based neural operator learning frameworks, we introduce an additional COMPOL aggregates across processes before applying positional attention.

**Per-process latent feature aggregation (COMPOL step)** Given multiple processes $m = 1, \ldots, M$, latent representations at layer $l-1$ are $v_{l-1}^m(x)$. First, compute aggregated latent representation across processes via attention:

$$z_{l-1}(x) = \mathcal{A}\left(v_{l-1}^1(x), v_{l-1}^2(x), \ldots, v_{l-1}^M(x)\right).$$

**Positional Attention enhanced by aggregated latent representation** Next, perform standard positional transformer attention at position $x_i$, but now augment it with the aggregated latent representation $z_{l-1}(x_i)$:

- Compute positional queries, keys, values:

$$Q_i' = Q'(v_{l-1}(x_i), z_{l-1}(x_i)), K_j' = K'(v_{l-1}(x_j), z_{l-1}(x_j)), V_j' = V'(v_{l-1}(x_j), z_{l-1}(x_j)).$$

- Apply enhanced positional attention:

$$v_l(x_i) = \sum_j \alpha_{ij}' V_j', \quad \text{where} \quad \alpha_{ij}' = \frac{\exp\left(Q_i' \cdot K_j'/\sqrt{d_k'}\right)}{\sum_j \exp\left(Q_i' \cdot K_j'/\sqrt{d_k'}\right)}.$$

Here, the latent aggregation $z_{l-1}(x_i)$ explicitly modulates the positional attention, guiding the attention mechanism to better capture multi-physics process interactions.

## C   Details of Synthetic Multi-Physics Datasets

**1-D Lotka-Volterra Equation**   The 1-D reaction-diffusion Lotka-Volterra system models predator-prey population dynamics through coupled partial differential equations:

$$\begin{cases} \frac{\partial u}{\partial t} = D_u \nabla^2 u + au - buv \\ \frac{\partial v}{\partial t} = D_v \nabla^2 v + cuv - dv \end{cases}$$

The system combines spatial diffusion (coefficients $D_u$ and $D_v$) with population interactions through reaction terms, generating complex spatio-temporal patterns. We study this system using a one-dimensional Gaussian Random Field initialization (length-scale $l = 0.1$, amplitude $\sigma = 1$) with periodic boundary conditions and uniform interaction parameters ($a = b = c = d = 0.01$) and uniform diffusion coefficients ($D_u = D_v = 0.01$) to examine the fundamental dynamics.

**1-D Belousov-Zhabotinsky Equations**   The Belousov-Zhabotinsky (BZ) reaction is a classical example of nonlinear chemical oscillations and pattern formation in reaction-diffusion systems (Taylor, 2002). The coupled BZ equations describe a reaction-diffusion process with three reactants $u(x,t)$, $v(x,t)$, and $w(x,t)$:

$$\begin{cases} \frac{\partial u}{\partial t} = \epsilon_1 \nabla^2 u + u + v - uv - u^2 \\ \frac{\partial v}{\partial t} = \epsilon_2 \nabla^2 v + w - v - uv \\ \frac{\partial w}{\partial t} = \epsilon_3 \nabla^2 w + u - w \end{cases}$$

where we set $\epsilon_1 = 1 \times 10^{-2}$, $\epsilon_2 = 1 \times 10^{-2}$, and $\epsilon_3 = 5 \times 10^{-3}$ and simulated the system from $t = 0$ to $t = 0.5$. The system exhibits complex spatiotemporal patterns due to the nonlinear coupling between the chemical species through terms like $uv$ and $u^2$. Initial conditions $u(x,0)$, $v(x,0)$, and $w(x,0)$ were generated using 1D Gaussian Random Field of Gaussian kernel and periodic boundary with correlation length being $0.03$ respectively. The equations were solved numerically using the fourth-order exponential time-differencing Runge-Kutta method (ETDRK4) with a resolution of 1024, subsequently subsampled to create datasets with resolution 256.

**2-D Grey-Scott Equation**   The Gray-Scott equations model pattern formation in chemical reactions through coupled partial differential equations:

$$\begin{cases} \frac{\partial u}{\partial t} = D_u \nabla^2 u - uv^2 + F(1 - u) \\ \frac{\partial v}{\partial t} = D_v \nabla^2 v + uv^2 + (F + k)v \end{cases}$$

The system tracks two chemical species: an activator ($u$) and an inhibitor ($v$). Their evolution is governed by diffusion (coefficients $D_u$ and $D_v$), an autocatalytic reaction ($uv^2$), and regulatory mechanisms through feeding rate $F$ and removal rate $k$. The interplay of these processes generates diverse patterns like spots and stripes, with their characteristics determined by the system parameters. To study these pattern-forming dynamics, we examine the system's evolution from a two-dimensional Gaussian Random Field initial condition to time $T = 20$, observing how perturbations develop into organized spatial structures. For all experiments reported, we employed the Gray-Scott system parameters $D_u = 0.12$, $D_v = 0.06$, $F = 0.054$, and $k = 0.063$.

# D  Details of Multiphase Flow Problem

Multiphase flow describes fluid mixtures moving simultaneously —a process critical for underground resource management, including oil and gas extraction, geological carbon storage, and nuclear waste disposal. We study oil-water two-phase flow using GEOS[1] on a 2D domain with water injection and oil extraction points on opposite boundaries. The domain permeability ranges from $1mD$ to $1000mD$, sampled based on the fractal distribution (Tang et al., 2021). Our dataset includes 1024 scenarios with varying boundary configurations, each simulated for 15 timesteps over $7.5 \times 10^6$ seconds. The goal is to predict the evolution of phase pressure ($P_p$) and saturation ($S_p$) distributions. Section D provides detailed specifications.

Multiphase flow is the simultaneous movement of two or more phases which is one of the most dominant subsurface processes for oil and gas extraction, transport of pollutants in subsurface environments, and geological carbon storage. Due to the presence of multiple phases, considerable complications are always encountered in describing and quantifying the nature of the flow. Here we present a 2D multiphase case of an underground oil-water two-phase flow. The components $\alpha$ (water) and $\beta$ (oil) of the multiphase flow fulfill the mass conservation equation:

$$\begin{cases} \frac{\partial M^\alpha}{\partial t} = -\nabla \cdot (\mathbf{F}_a^\alpha + \mathbf{F}_d^\alpha) + q^\alpha \\ \frac{\partial M^\beta}{\partial t} = -\nabla \cdot \left( \mathbf{F}_a^\beta + \mathbf{F}_d^\beta \right) + q^\beta \end{cases}$$

where $\mathbf{F}_a^\alpha$ and $\mathbf{F}_a^\beta$ are the advective mass flux, $\mathbf{F}_d^\alpha$ and $\mathbf{F}_d^\beta$ are the diffusive mass flux, $q^\alpha$ and $q^\beta$ are the source or sink terms, and $\mathbf{M}^\alpha$ and $\mathbf{M}^\beta$ are the mass accumulation terms given by

$$\begin{cases} M^\alpha = \phi \sum_p S_p \rho_p X_p^\alpha \\ M^\beta = \phi \sum_p S_p \rho_p X_p^\beta \end{cases}$$

In the mass accumulation terms, $\phi$ is the porosity, $S_p$ is the saturation of phase $p$, $X_p^\alpha$ or $X_p^\beta$ is the mass fraction of component $\alpha$ or $\beta$ in phase $p$, and $\rho_p$ is the density of phase $p$. In the scenarios of subsurface oil-water two-phase flow, $\mathbf{F}_d^\alpha$ and $\mathbf{F}_d^\beta$ including molecular diffusion and hydrodynamic dispersion are often negligible when compared to $\mathbf{F}_a^\alpha$ and $\mathbf{F}_a^\beta$. For simplicity, we don't include diffusion terms in our simulations. And the advective mass flux equations of component $\alpha$ and $\beta$ are

$$\begin{cases} \mathbf{F}^\alpha\big|_a = \sum_p X_p^\alpha \rho_p \mathbf{u}_p \\ \mathbf{F}^\beta\big|_a = \sum_p X_p^\beta \rho_p \mathbf{u}_p \end{cases}$$

Here, $\mathbf{u}_p$ is the Darcy velocity of phase $p$ defined as follows:

$$\mathbf{u}_p = -k \left( \nabla P_p - \rho_p \mathbf{g} \right) k_{rp} / \mu_p$$

where $k$ is the absolute permeability tensor, $P_p$ is the fluid pressure of phase $p$, $\mathbf{g}$ is the gravitational acceleration, $k_{rp}$ is the relative permeability of phase $p$, and $\mu_p$ is the viscosity of phase $p$. The fluid pressure for wetting phase $P_w$ or non-wetting phase $P_n$ is

$$P_n = P_w + P_c$$

where $P_c$ is the capillary pressure.

In this study, the simulation is performed on GEOS which is an open-source multiphysics simulator. To simulate the oil-water two-phase flow, water is injected into a 2D ($64 \times 64$ grids) domain from two randomly placed sources on the left boundary. And at the same, oil is produced from two randomly placed sinks on the right boundary. The permeability fields ($k_x = k_y$) are generated using a fractal algorithm with $k_{min} = 1mD$, $k_{max} = 1000mD$, and $k_{base} = 100mD$. And the porosity field is generated by the following correlation:

$$\phi = \left( \frac{k_x}{1 \times 10^{-15} \times 0.0009} \right)^{\frac{1}{4.0001}} \times \frac{1}{100}$$

For the 1024 cases in the dataset, we keep all other conditions identical and only change the locations of sources and sinks. For each case, the multiphysics solver is performed every $5 \times 10^5$ seconds for 15 times and the whole simulation has a time duration of $7.5 \times 10^6$. During the simulation, the spatial and temporal results of phase pressure and phase saturation are stored at every time step which makes the dimension of the dataset to be $1024 \times 15 \times 64 \times 64 \times 2$. Our objective here is to learn the mapping from an earlier spatial distribution of $S_p$ and $P_p$ to that of a later time step.

---

[1]https://github.com/GEOS-DEV/GEOS

## E   Details of Thermo-Hydrolic-Mechanics Problem

The coupled thermo-hydro-mechanical (THM) processes in porous media and fractured rock are associated with a wide range of applications including geothermal energy extraction, induced seismicity from fluid injection, reservoir stimulation, and nuclear water disposal in subsurface. All the problems involve strong coupling among pressure diffusion, heat transfer, and change of in-situ stress and rock deformation. Here we present a 2D THM case adapted from the 1D thermo-hydro-mechanical problem presented in Gao & Ghassemi (2020). The governing equations (momentum balance equation, fluid mass balance equation, and heat transfer equation) of THM problems can be derived from the thermo-poroelasticity theory of porous and permeable rock.

$$
\begin{aligned}
& G u_{i,jj} + \frac{G}{1-2\nu} u_{j,ji} - \alpha p_{,i} - \frac{2G\alpha_m^T(1+\nu)}{3(1-2\nu)} T_i + F_i = 0, \\
& \frac{\partial p}{\partial t} = M\left[\frac{k}{\mu}p_{jj} - \alpha\frac{\partial \varepsilon_{kk}}{\partial t} + \left(\alpha\alpha_m^T + \phi_0(\alpha_f^T - \alpha_m^T)\right)\frac{\partial T}{\partial t}\right] + \gamma, \\
& \frac{\partial T}{\partial t} = \frac{1}{\rho_t C_t}(\kappa^T T_i)_i - \frac{\rho_f C_f}{\rho_t C_t} v_i T_i.
\end{aligned} \tag{2}
$$

where $G$ is the shear modulus of the solid skeleton, $\nu$ is the Poisson's ratio, $\alpha$ is the Biot's coefficient, $p_i$ is the pore pressure gradient on $i$ direction, $\alpha_m^T$ is the thermal expansion coefficient of the porous media, $T_i$ is the temperature gradient on $i$ direction ,$F_i$ is the body force term on $i$ direction, $\frac{\partial p}{\partial t}$ is the time change rate of pore pressure, $M$ is the Biot modulus, $\frac{k}{\mu}$ is the hydraulic conductivity, $\frac{\partial \varepsilon_{kk}}{\partial t}$ is the time rate change of volumetric strain, $\phi_0$ is the initial porosity, $\alpha_f^T$ is the thermal expansion coefficient of the fluid, $\frac{\partial T}{\partial t}$ is the time rate change of temperature, $\gamma$ is the source or sink term of the fluid, $\rho_t$ is the total density of the porous media, $C_t$ is the total heat capacity, $\kappa^T$ is the thermal conductivity tensor, $\rho_f$ is the fluid density, $C_f$ is the fluid heat capacity, and $v_i$ is the fluid velocity component.

In this study, the simulation is performed on GEOS which is an open source multiphysics simulator. To simulate pressure diffusion, heat transfer, and change of in-situ stress and rock deformation, a 2D ($64 \times 64$ grids) domain on XY plane with actual size of 32 m by 32 m and thickness of 0.5 m in Z direction is employed. For each simulation, a random temperature field with a range of 273 to 323 K and random permeability fields are mapped to the grid. The permeability fields ($k_x = k_y$) are generated using a fractal algorithm with $k_{min} = 10\,D$, $k_{max} = 200\,D$, and $k_{base} = 50\,D$.

For the 1250 cases in the dataset, we keep all other conditions identical and only change the realizations of temperature and permeability fields. For each case, the multiphysics solver is performed 12 times as $2 \times 500$ seconds, $4 \times 1250$ seconds, $4 \times 2250$ seconds, and $2 \times 4000$ seconds consecutively and the whole simulation has a time duration of $2.3 \times 10^4$ seconds. During the simulation, the spatial and temporal results of pore pressure, strain, and temperature are stored at every time step which makes the dimension of the dataset to be $1250 \times 12 \times 64 \times 64 \times 3$. Our objective here is to learn the mapping from an earlier spatial distribution of pore pressure, strain, and temperature to that of a later time step.

## F Predictions of COMPOL and Baseline Models vs. Ground Truth

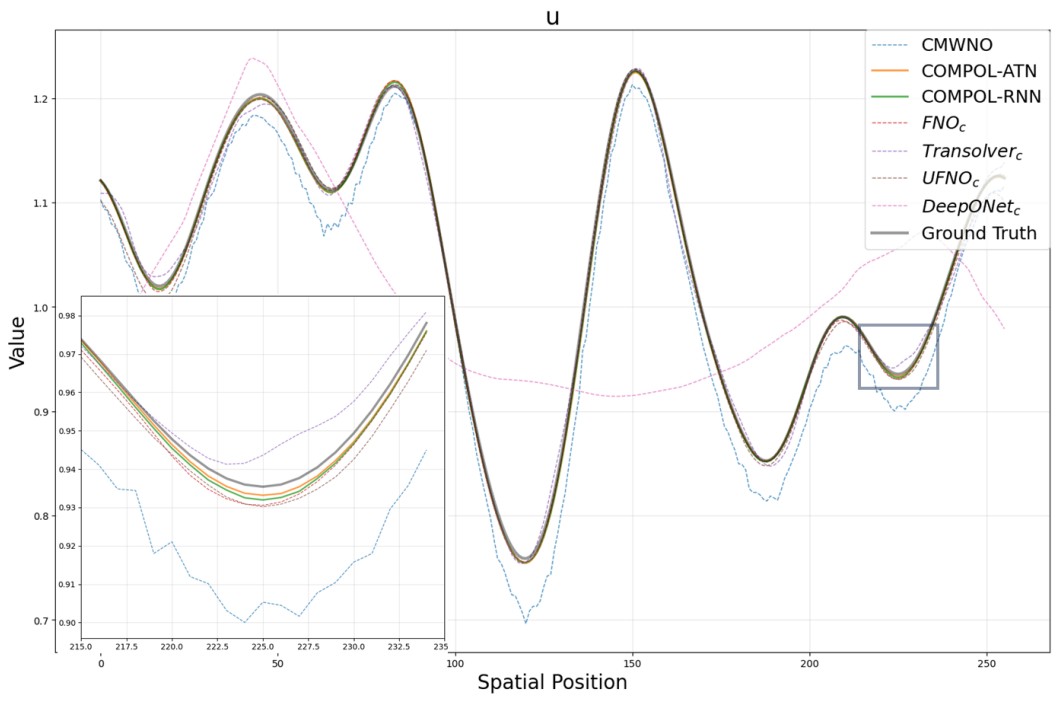

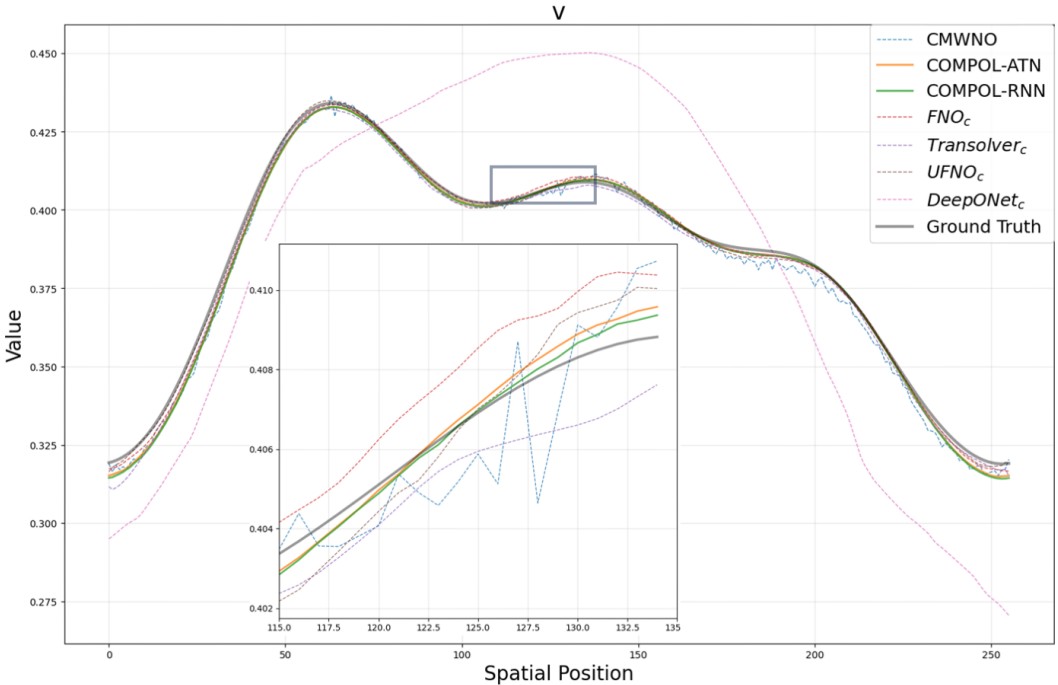

Figure 4: Predictions of COMPOL and Baseline Models vs. Ground Truth of Lotka-Volterra Using 512 Training Samples

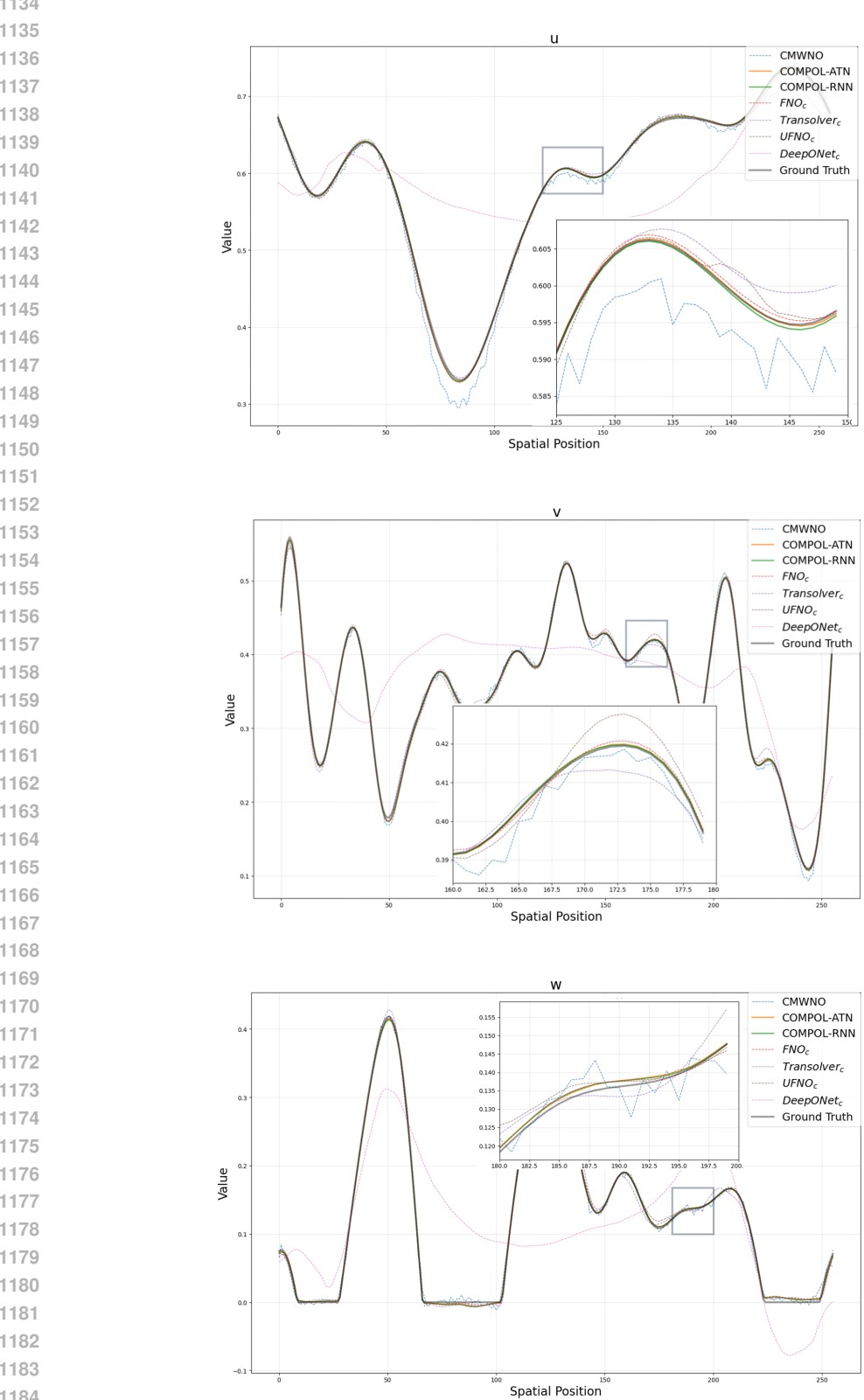

Figure 5: Predictions of COMPOL and Baseline Models vs. Ground Truth of Belousov-Zhabotinsky Using 512 Training Samples

# G  A Comprehensive Investigation of COMPOL's Coupling Mechanisms

According to Figure 2, our proposed model consists of three key components: (1) $c_{l-1}$ compresses and aggregates outputs from multiple processes, denoted as $\mathbf{v}_{l-1}^p$ ($p = 1, ..., M$), at the current layer; (2) $a_{l-1}$ aggregates latent representations from all previous layers, $\mathcal{V}_{l-1}$; and (3) $b_{l-1}$ integrates the aggregated information back into the original process-specific outputs $\mathbf{v}_{l-1}^p$. The default implementations for these components are linear transformation for $c_{l-1}$, attention mechanisms for $a_{l-1}$, and simple addition for $b_{l-1}$. Alternative configurations tested include: replacing linear transformation in $c_{l-1}$ with addition; substituting the attention mechanism in $a_{l-1}$ with ResNet-style skip connections, RNN, or multihead attention; and using concatenation followed by dimensionality reduction instead of addition for $b_{l-1}$. We particularly examined alternative methods for $a_{l-1}$ to explore how attention compares against less expressive methods like RNNs or ResNet-like skip connections, which inherently discard older information from layers further back.

We also explored the impact of employing separate Fourier Neural Operators (FNOs) for each process versus a single FNO treating all processes as channels. Additionally, we varied the model width parameter to investigate how performance scales with the number of parameters, limited to one-dimensional cases due to computational constraints.

Figures 7 to 10 summarize performance evaluations of 160 models. Dot labels indicate the width parameter, with negative values representing fewer parameters and positive values indicating increased parameters. Standard deviations are shown by error bars. Figure 7 reveals a general trend where separate FNOs with linear transformation outperform those with additive mixing, single FNO models, and separate FNOs without inter-process communication. Test error versus parameter percentage demonstrates a U-shaped curve, with optimal performance around 90%. In Figure 8, attention and RNN aggregation significantly decrease test errors in separate FNO scenarios but adversely impact performance in single FNO setups. Our model consistently achieves superior results across various parameter settings, particularly with attention mechanisms. Differences between additive and concatenative approaches in $b_{l-1}$ are minimal. Figure 9 shows a clear scaling law, as logarithmic training errors decrease linearly with logarithmic parameter increases. Figure 10 indicates that overfitting occurs as parameter counts become large. Notably, separate FNOs without inter-process communication perform poorly, emphasizing the critical role of effective information mixing.

Similar trends are evident in results from the Belousov-Zhabotinsky equations (Figures 11–14). Additionally, Figure 15 demonstrates that even for single-process scenarios like the 1D Burgers equation, our model's ResNet-like architecture significantly enhances performance by enabling deeper layers to learn residual corrections rather than entire solutions.

We further evaluated model performance on deliberately unrelated processes by concatenating distinct channels from Belousov-Zhabotinsky, Lotka-Volterra, and Burgers equations. Figures 17 and 16 indicate that standard FNOs perform poorly in this setting due to inefficient interchannel communication in the frequency domain. In contrast, our model, based on separate FNOs with controlled inter-process communication in the spatial-temporal domain, performs robustly, matching or slightly surpassing separate FNO performance. This advantage highlights our model's capability to flexibly manage both strongly and weakly related processes without requiring prior knowledge of their interplay, making it highly suitable for fully data-driven scenarios.

In conclusion, the ablation study consistently demonstrates that our proposed architecture outperforms alternative configurations under equivalent parameter budgets. Modifying or omitting components generally leads to performance degradation, underscoring the effectiveness of the integrated design.

Less channel/process mixing (between different processes) →

Less feature aggregation (between different Fourier Layers' output) ↓

| Feature aggregation | Channel mixing | Whole FNO for all processes | Separate FNOs for different processes | | | | No channel/process mixing |
|---|---|---|---|---|---|---|---|
| | | | With channel/process mixing | | | | |
| | | | Linear transformation | | Add (less channel mixing) | | |
| | | | Feature aggregation to next layer input | | Feature aggregation to next layer input | | |
| | | | Add | Concat | Add | Concat | |
| With feature aggregation | **Multihead attention** | | fno1d_multi_atten_add | fno1d_multi_atten_concat | fno1d_multi_atten_add_less_channel_mixing | fno1d_multi_atten_concat_less_channel_mixing | |
| | **Attention** | fno1d_whole_attention | **fno1d_atten_add** | fno1d_atten_concat | fno1d_atten_add_less_channel_mixing | fno1d_atten_concat_less_channel_mixing | fno1d_separate_attention |
| | **RNN** | fno1d_whole_rnn | fno1d_rnn_add | fno1d_rnn_concat | fno1d_rnn_add_less_channel_mixing | fno1d_rnn_concat_less_channel_mixing | fno1d_separate_rnn |
| | **Resnet** | fno1d_whole_resnet | fno1d_resnet_add | fno1d_resnet_concat | fno1d_resnet_add_less_channel_mixing | fno1d_resnet_concat_less_channel_mixing | fno1d_separate_resnet |
| **No feature aggregation** | | fno1d_concat | | | | | fno1d_separate |

Figure 6: Models tested in the ablation study arranged by two dimensions: feature aggregation of layers and channel/process mixing. fno1d_atten_add is the model we use by default

Figure 7: Test Errors vs Percentage of FNO Parameters of Lotka-Volterra Using 512 Training Samples

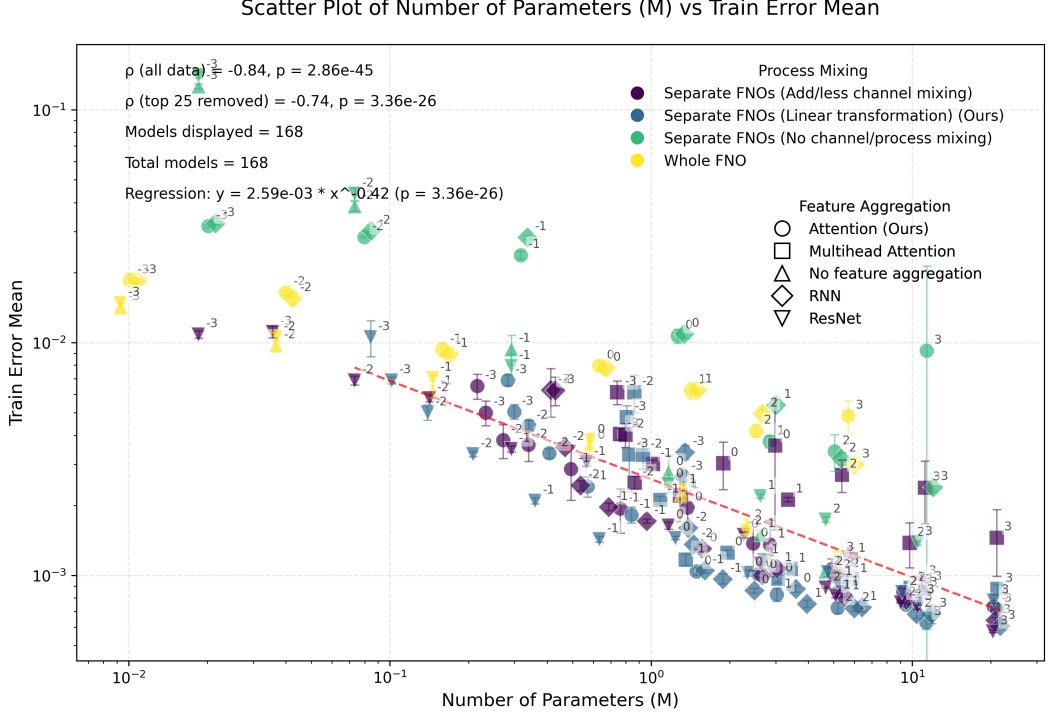

Figure 8: Test Errors vs Number of Parameters of Lotka-Volterra Using 512 Training Samples

Figure 9: Test Errors vs Number of Parameters of Lotka-Volterra Using 512 Training Samples

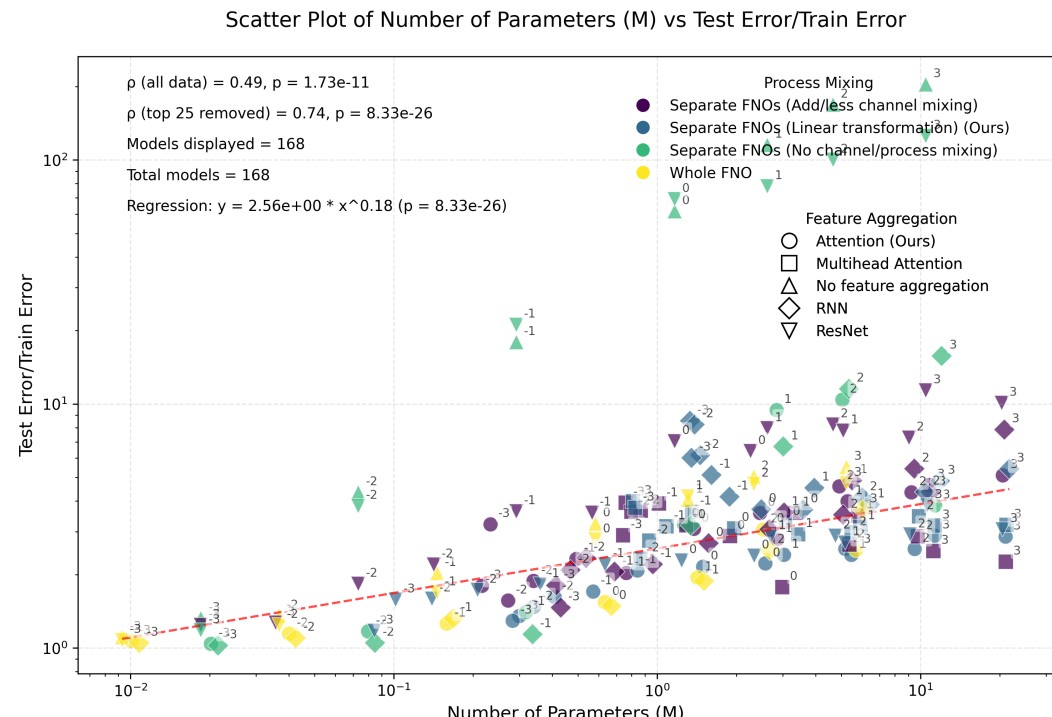

Figure 10: Test Errors Over Train Errors Ratio vs Number of Parameters of Lotka-Volterra Using 512 Training Samples

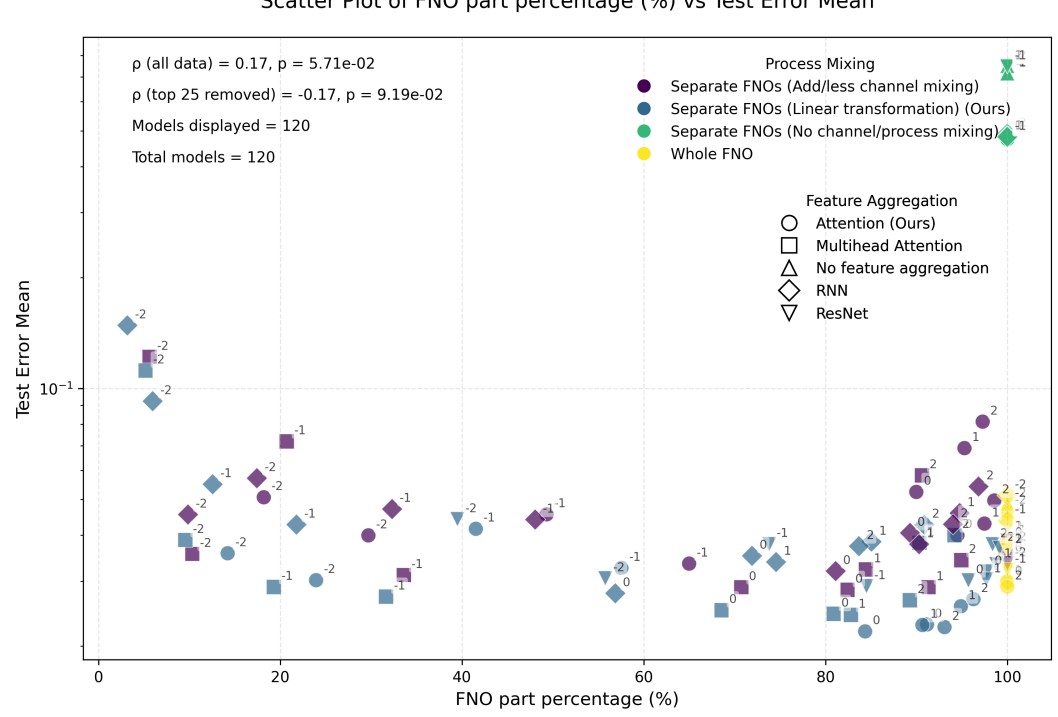

Figure 11: Test Errors vs Percentage of FNO Parameters of Belousov-Zhabotinsky Using 512 Training Samples

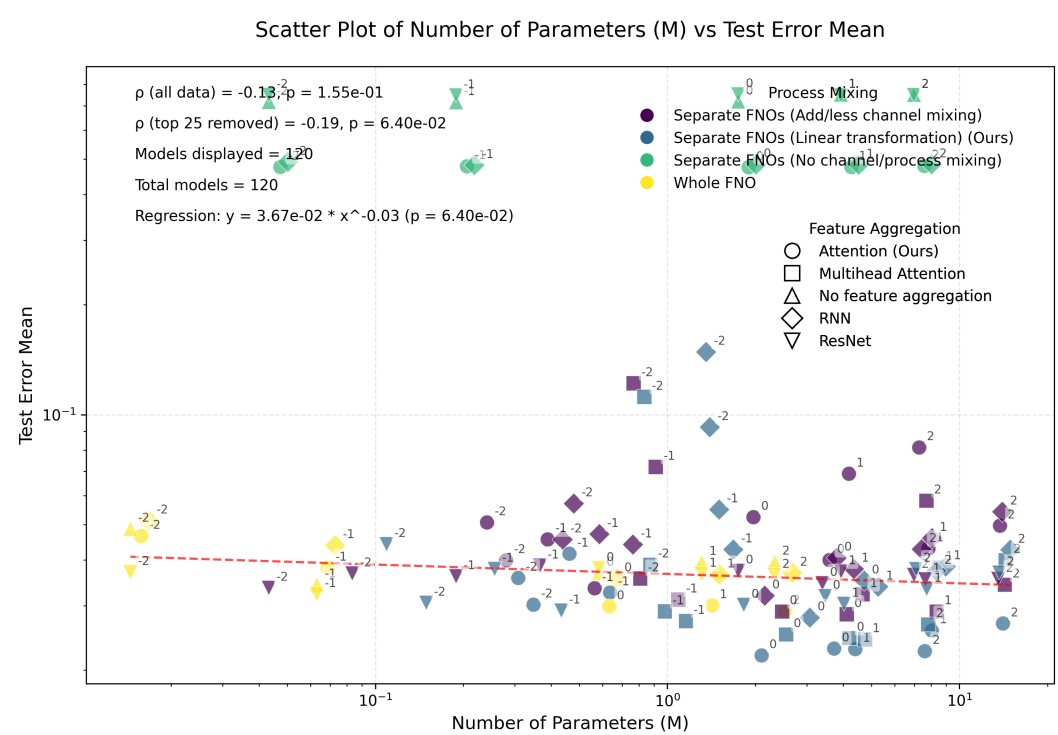

Figure 12: Test Errors vs Number of Parameters of Belousov-Zhabotinsky Using 512 Training Samples

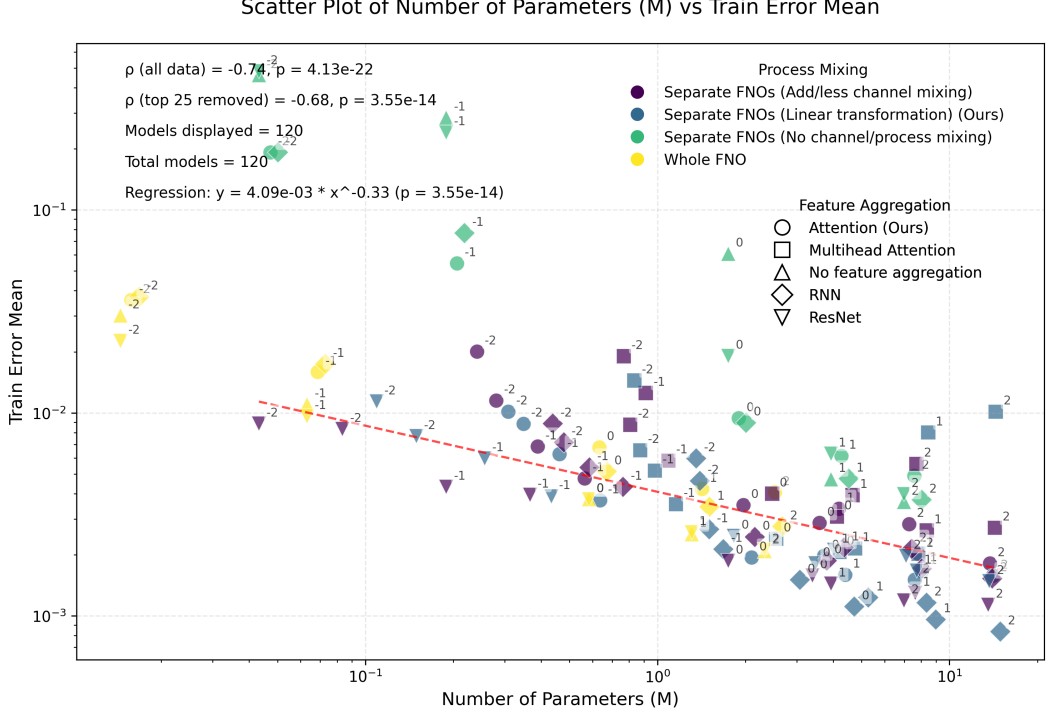

Figure 13: Test Errors vs Number of Parameters of Belousov-Zhabotinsky Using 512 Training Samples

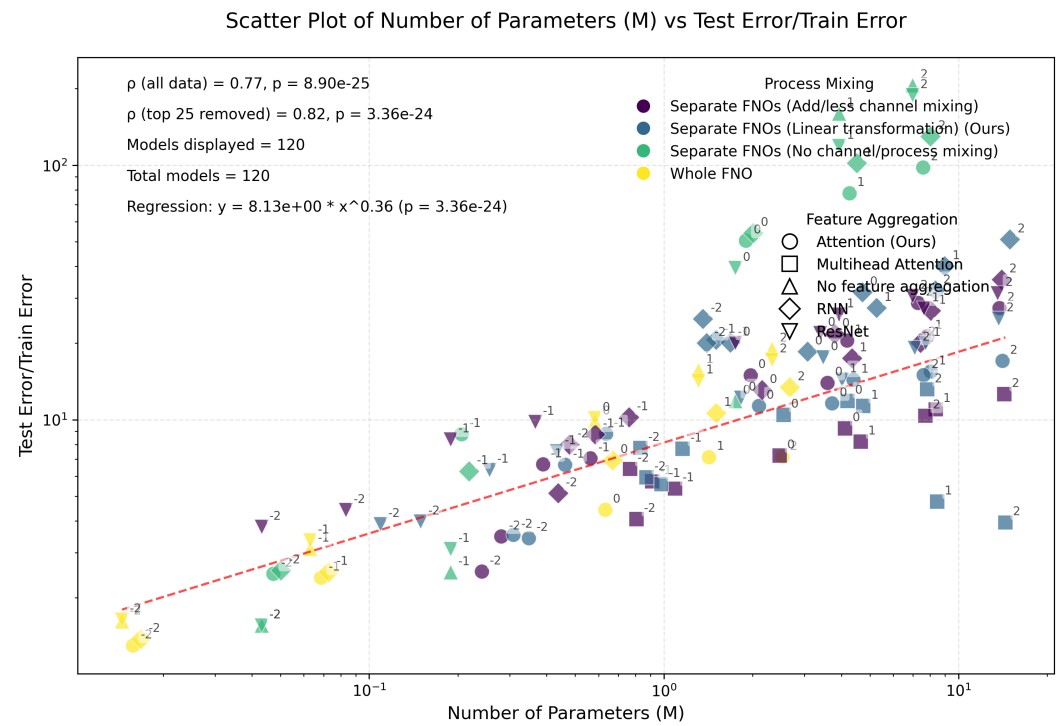

Figure 14: Test Errors Over Train Errors Ratio vs Number of Parameters of Belousov-Zhabotinsky Using 512 Training Samples

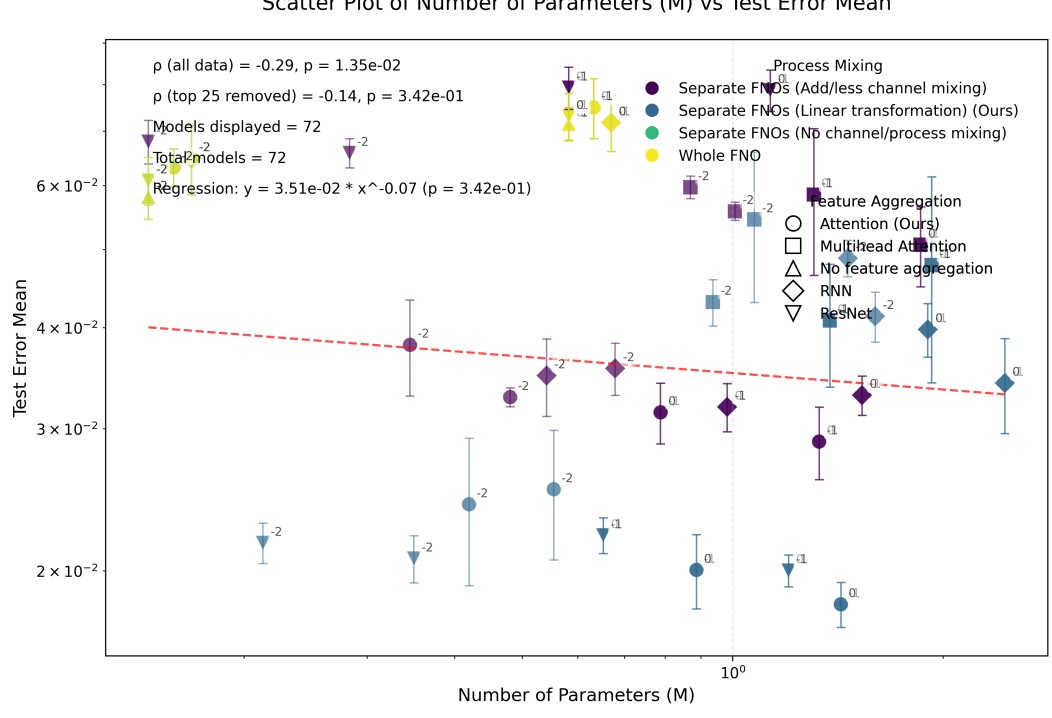

Figure 15: Test Errors vs Number of Parameters of Burgers Using 512 Training Samples

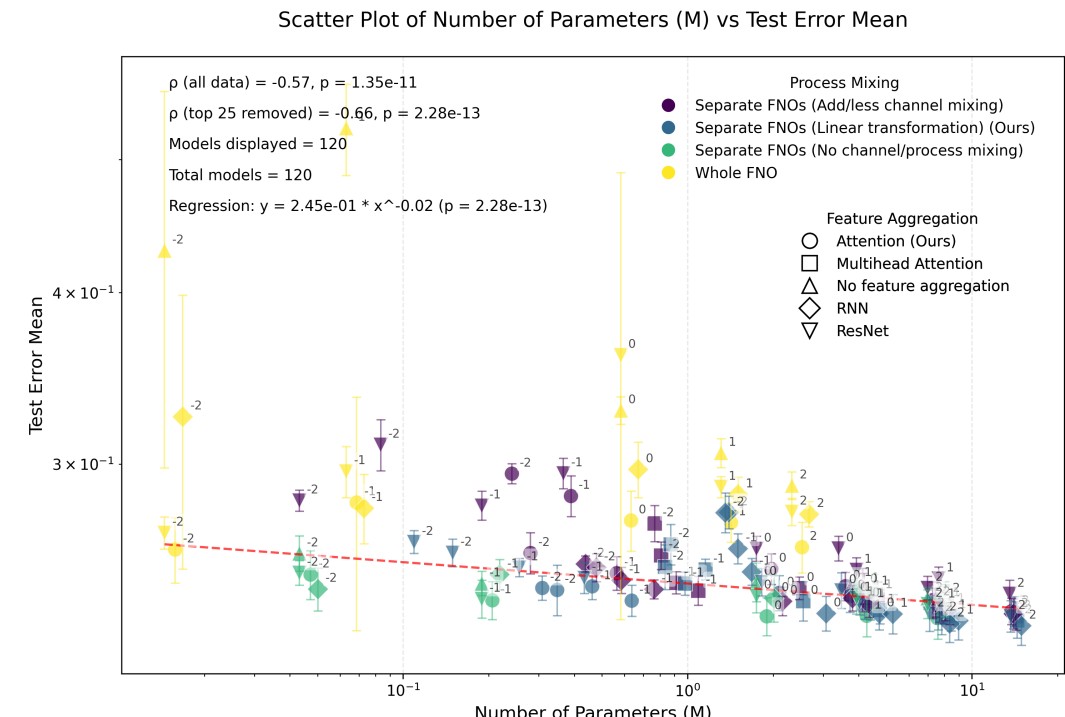

Figure 16: Test Errors vs Number of Parameters of Our Synthetic Test Case from BZ, LV and Burgers Using 512 Training Samples

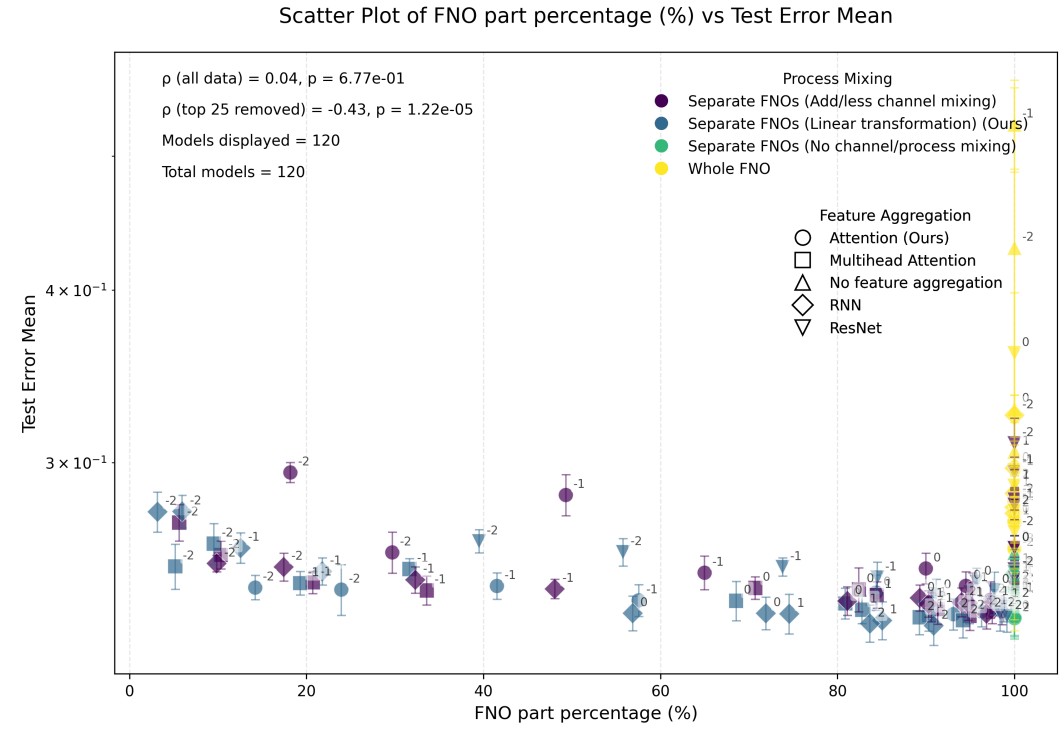

Figure 17: Test Errors vs Percentage of FNO Parameters of Our Synthetic Test Case from BZ, LV and Burgers Using 512 Training Samples

