# OpenReview forum: "COMPOL: A Unified Neural Operator Framework for Scalable Multi-Physics Simulations"
_ICLR.cc/2026/Conference — ICLR 2026 Conference Withdrawn Submission_

### Official Review · Reviewer_RALb · 2025-10-19

**Soundness:** 2
**Presentation:** 2
**Contribution:** 2
**Rating:** 2
**Confidence:** 4

**Summary:**

This paper introduces COMPOL, a neural operator framework aimed at modeling coupled multi-physics PDE systems. The key idea is to extend the Fourier Neural Operator (FNO) with additional latent feature aggregation mechanisms based on recurrent (GRU) and attention modules. The authors claim that COMPOL is architecture-agnostic and capable of generalizing to other operator learning paradigms (e.g., DeepONet, GNO, transformer-based solvers). The paper includes theoretical analyses (universal approximation, convergence, stability) and experimental evaluations on several multi-physics datasets, reporting improved predictive accuracy and generalization over baseline neural operators.

**Strengths:**

- Addresses an important problem: scalable multi-physics operator learning
- Incorporates latent aggregation ideas that could, in principle, enhance cross-process information flow.
- Provides a reasonably thorough set of experiments and ablation studies.
- Includes theoretical justification for stability and approximation, even if standard.

**Weaknesses:**

1. Clarity and Writing Quality
    - The paper is difficult to follow, especially in Section 3.2–3.3. The presentation is redundant, vague, and hand-waving.
    - Key terms such as “process-specific evolution,” “inter-process interactions” and “global context” are not rigorously defined.

2. Lack of Conceptual Novelty
     - Despite being described as a new framework, COMPOL is largely built upon FNO, with some modifications—essentially adding a latent GRU or attention module between  FNO layers.
     - The claimed unified or architecture-agnostic nature is not demonstrated; all experiments are conducted with FNO backbones only.

3.  Insufficient Benchmarking
       - Despite the availability of public multi-physics benchmarks(*PDEArena*, *PDEBench*, *the Well dataset*), the authors rely solely on custom datasets.
       - Stronger recent baselines such as DPOT, MPP, PDE-Transformer, and CViT are not included.
       - The main baselines (FNO, UFNO, Transolver, DeepONet) are either outdated or adapted in ways that may not reflect their best performance.

4.  Theoretical Part is Superficial
       - The so-called “proofs” of universal approximation, convergence, and stability are directly adapted from existing neural operator theorems (Kovachki et al., 2023) and do not depend on the new aggregation mechanism.  Hence, the theoretical contribution is minimal and not novel.

**Questions:**

1. Would a self-attention layer in a standard transformer not already perform the proposed “cross-process aggregation”? How does COMPOL differ fundamentally from multi-head attention across channels?

2.  If COMPOL is truly architecture-agnostic, why are all experiments conducted on FNO backbones? Can the authors show results with DeepONet or transformer-based operators?

3. What happens when the number of coupled processes increases significantly (e.g., >10)? Is the recurrent aggregation stable and scalable?

4. Are the reported performance gains still evident when parameter count and compute budget are matched with strong baselines (e.g., models scaled to ~100M parameters) under identical training conditions?

---

### Official Review · Reviewer_T1Kp · 2025-10-26

**Soundness:** 2
**Presentation:** 2
**Contribution:** 2
**Rating:** 2
**Confidence:** 4

**Summary:**

The paper proposes the following approach to handle coupled multi-variable PDEs:
1) Use a separate stack of neural operator layers per variable
2) At each layer, aggregate information across all variables via attention or GRU to create a shared representation z_l
3) Feed this aggregated z_l back into each variable's neural operator at the next layer

The approach is validated against baselines on multiple benchmarks and consistently demonstrates improvement, though the experimental setup raises concerns due to the lack of established benchmarks, as all the datasets are generated by the authors themselves.

**Strengths:**

**Originality**
- The proposed approach is a solid engineering contribution: using separate neural operator streams per variable with learned aggregation (GRU/attention) is a reasonable architectural choice for coupled PDE systems.
- The idea of maintaining separate streams per variable could be valuable in high-scale distributed computing regimes, where each stream could operate on a separate device.
- The framework is architecture-agnostic and generalizes beyond FNO (extensions to DeepONet, GNO, LNO, and Transolver shown in Appendix B).

**Quality**
- Comprehensive benchmarking across diverse physical systems (biological reaction-diffusion, chemical reactions, geological multiphase flows, and thermo-hydro-mechanical processes)
- Extensive ablation study in Appendix G analyzing approx. 160 model configurations across different design choices
- Authors provide detailed hyperparameters, training procedures, dataset generation methods, and PDE specifications in appendices

**Clarity**
- Well-structured supplementary material with theoretical analysis, implementation details, and additional experiments

**Significance**
- The proposed method demonstrates consistent improvements across multiple applications, indicating practical merit.
- It also shows that treating coupled PDEs with separate-but-communicating operator stacks works better than naive channel concatenation, which is a useful insight.

**Weaknesses:**

My main issue with the paper is that it exaggerates the novelty and tries to present the framework as something it is not.

- I find the "multi-physics" framing misleading: from a deep learning point of view, what is the difference between a multi-channel problem and a multi-physics problem? It seems to me that the problem setup is simply multi-channel input, multi-channel output. Furthermore, when the authors make a distinction between multi-physics and multi-channel operator learning, I do not buy it. I believe that framing the contribution as neural operators for coupled multi-variable PDEs would be somewhat more honest.

- A major concern is that the paper tries to appear more theory-grounded and rigorous than it really is. I would suggest moving the theoretical claims to the supplementary material altogether and using the space for clearer notation definitions and detailed framework descriptions. My issues with the current framing are as follows:
  - Theorem A.1 (Universal Approximation) just cites existing universal approximation results for NOs. I do not see how COMPOL is theoretically grounded in the theorem.
  - Theorem A.2 (Convergence via Fixed-Point) is fundamentally misapplied. The theorem claims COMPOL's layer-wise updates "converge to a unique fixed point" via the Banach fixed-point theorem. However, COMPOL is a finite-depth feedforward network (with 4 layers), not an iterative procedure. By this logic, any deep network would have convergence guarantees, but that is not the case. The Banach fixed-point theorem applies to iterative methods (optimization, equilibrium models), not finite forward passes.
  - Theorem A.3 (Stability) holds, but it just states that a composition of Lipschitz functions is Lipschitz. I am not sure if that makes COMPOL any more rigorous.

- The experimental setup in Section 5.1 is questionable for three reasons.
  - First, there are the datasets themselves. Despite the field having multiple benchmarks for the specific "multi-physics" setup that were used in other papers, the authors generate all the data themselves and do not use any established benchmarks. This is problematic for both the authors (it is hard to validate proper implementation for a baseline without a ground truth reference) and readers (it is not clear whether those datasets provide meaningful evidence). I strongly suggest the authors use at least one dataset that was previously used by other methods (e.g., CMWNO) or an established benchmark (such as Multiphysics Bench [1]). That would provide a standardized benchmark with established results and make their experimental claims much stronger, in my opinion.
  - Second, COMPOL uses a stack of neural operators, while other models do not. Hence, it has M times more parameters, which is likely a major reason behind the large performance gap. When the parameter count is matched, the gap shrinks (Table 5.2), although it still remains significant. I would suggest that the authors clarify the parameter count in Table 1, and if it is significantly different across models, redo the experiments matching the models' capacity.
  - Third, CMWNO, the only baseline actually designed for coupled PDEs, is barely tested and performs suspiciously poorly. In the CMWNO paper, Table 1, the authors compare against FNO on the Gray–Scott benchmark and perform significantly better. Furthermore, I do not see why the result on Gray–Scott is not reported in this paper. On a similar note, CMWNO is originally benchmarked on the Belousov–Zhabotinsky dataset, where it performs significantly better than FNO, while in this paper, FNO outperforms it significantly. Where does the discrepancy come from? I suggest that the authors conduct their experiments on the same dataset as in the CMWNO paper, given that data generation is provided for a fair comparison. I have a strong suspicion right now that there is a bug in the implementation the authors use, which, again, using an original dataset from the CMWNO paper would help to clarify.

- I find the presentation imprecise and confusing.
   - For example, in Section 3.3: "V_l = {{v^m_j}^M_{m=1}}^l_{j=0}" suggests aggregating over ALL previous layers explicitly, but GRU only sees the current layer and previous hidden state.
    - It is not clear how the aggregation happens. Is it per spatial point? If so, it would feel like modern CNNs that have depthwise/pointwise convolution separation, which is actually nice.
    - How is z_l "fed back" to the FNO layers -- through concatenation, addition, or something else?
    - "Recurrent aggregation" implies temporal recurrence, but it's layer-wise aggregation, so it feels more like a fancier way of implementing residual connections. Wouldn't the latter work just as well?

- The language used across the paper is somewhat ambiguous and imprecise. For example, words like "sophisticated" or "intricate" -- I do not think they fit scientific language. I suggest polishing the manuscript to eliminate such words.

[1] https://arxiv.org/abs/2505.17575

**Questions:**

The main limitation for me is the experimental setup. Would authors be open to do another set of experiments on either Multiphysics Bench, or datasets used in the CMWNO paper?

---

### Official Review · Reviewer_A2xK · 2025-10-30

**Soundness:** 3
**Presentation:** 3
**Contribution:** 2
**Rating:** 6
**Confidence:** 4

**Summary:**

This paper introduces a novel neural operator framework (COMPOL) for multi-physics PDE problems. COMPOL include a recurrent and an attention based aggregation maechanics to model interdependency across physics process. The experiments show superiors accuacy across multiple new PDE datasets.

**Strengths:**

- The paper considers a relatively new problem setting of multi-physics modeling.
- It provides several new datasets consist of multiple physics fields and variables.
- The architecture is flexible and agnostic to the specific inner neural operator choices.

**Weaknesses:**

1. The recurrent module and attention mechanism have been widely studied in ML and PDE community. Especially, the inter-physics attention has been applied in [1]. It will be helpful to discuss and compare with the previous work.
2. Using recurrent structure and attention usually increase the runtime significantly, it will be helpful to report the runtime and plot convergence of the cost accuracy tradeoff (at what runtime, the model get what error rate). This will be helpful to justify the extra runtime and memory usage.


[1] Rahman, Md Ashiqur, et al. "Pretraining codomain attention neural operators for solving multiphysics pdes." Advances in Neural Information Processing Systems 37 (2024): 104035-104064.

**Questions:**

1. It is an interesting question how the number of physics field correlated with the improvement of performance. Intuitive the aggregation is helpful when there is more physics coupled. For example, a standard NS equation has x-velocity, y-velocity, and p-pressure. How many physics field will the COMPOL model be helpful?
2. Most of the datasets studied are newly generated. It will be interested to have an existing dataset, especially on weather foreacst. In ERA5 there are over 90 physics fields (temperature, velocity, humidity, etc). The model should show improve of performance.

Minor suggestion: please make the main figure large (full linewidth) and add figures for each dataset.

---

### Official Review · Reviewer_S2o9 · 2025-11-01

**Soundness:** 3
**Presentation:** 2
**Contribution:** 3
**Rating:** 6
**Confidence:** 3

**Summary:**

COMPOL introduces a neural operator framework for coupled-multiphysics simulations. This framework is architecture agnostic, and extends classic neural operator models with latent aggregation mechanisms (Gated recurrent Unit based and Attention-based). This framework was tested on a range of PDE benchmarks, on which it outperforms other classic neural operator models.

**Strengths:**

-	Theoretical proof of convergence of latent updates and of the stability of the aggregation
-	State of the art performance on multiple datasets
-	Introduction of a novel, architecture agnostic framework, for tackling Multiphysics problems
-	Ablation studies showcase the importance of each component.

**Weaknesses:**

-	Lack of clarity in certain proofs
-	Lack of clarity sometimes in the writing. The architecture introduces with the two types of aggregation is complex, and Figure 2 does not help decipher it.
-	The text formatting is very dense; some equations and formulas would deserve to be highlighted in equation blocks.

**Questions:**

-	Could the authors expand the proof of Theorem A2? Also, should the condition be that L_{|Phi} + L_A L_{|Phi} < 1

---

### Note · Authors · 2026-02-26

I have read and agree with the venue's withdrawal policy on behalf of myself and my co-authors.

---

### Meta-Review · Area_Chair_mUGP · 2025-12-29

**Summary:**

I find this submission addresses a relevant problem—learning operators for coupled multi-physics PDE systems—with a reasonable architectural proposal: maintain separate neural operator streams per physical process and aggregate information across streams via GRU or attention mechanisms. The idea has merit, and the ablation study in Appendix G is thorough.

However, I am not convinced by the experimental evidence. The most concerning issue is the CMWNO baseline, which performs significantly worse here than in its original publication on the same Belousov-Zhabotinsky benchmark. Reviewer T1Kp correctly identifies that CMWNO significantly outperforms FNO on BZ and Gray-Scott in its original paper, yet here FNO beats CMWNO by a wide margin and Gray-Scott results for CMWNO are omitted entirely. This discrepancy suggests either implementation bugs or incomparable experimental conditions, and without clarification, I cannot trust the baseline comparisons.

Additionally, COMPOL uses M separate FNO stacks (one per process), giving it roughly M times the parameters of concatenation-based baselines. While Table 2 shows parameter-matched comparisons where COMPOL still leads, the headline results in Table 1 do not account for this disparity. Finally, all five benchmarks are custom-generated despite established multi-physics benchmarks existing (PDEBench, PDEArena, MultiphysicsBench). This prevents verification against known baseline performance and raises reproducibility concerns.

**Reviewer Concerns:**

Note that the authors did not provide a rebuttal and did not engage with the referees.

Reviewers T1Kp and RALb raised concerns about baseline implementation and experimental methodology. The CMWNO discrepancy was flagged specifically: in its original paper, CMWNO outperforms FNO on Belousov-Zhabotinsky, but here the relationship is reversed with no explanation. This concern was not addressed and remains outstanding. The parameter count issue was partially addressed via Table 2, which shows COMPOL still outperforms parameter-matched FNO variants, though the gap shrinks considerably. However, the main Table 1 comparisons remain unfair, and this was the basis for most claims.

Reviewers T1Kp and RALb also raised concerns about missing standard benchmarks and recent baselines. PDEBench, PDEArena, the Well dataset, and MultiphysicsBench were all mentioned as available alternatives that would provide fairer comparison ground. Methods like DPOT, MPP, and PDE-Transformer were noted as missing baselines. These concerns were not addressed.

Reviewer T1Kp raised a valid point about the theoretical claims being misapplied—specifically, that the Banach fixed-point theorem governs iterative convergence, not finite feedforward passes. Reviewers S2o9 and RALb noted the theoretical contributions were either unclear or superficial. These concerns were not addressed.

Reviewer A2xK noted that Rahman et al. (2024) already explored inter-physics attention for neural operators, reducing the novelty claim. Runtime comparisons were also requested. Neither concern was addressed.

**Reviewer Scores:**

Reviewer S2o9: 6 → 5. Their weak positive stance ("would not mind if paper is rejected") would likely shift after seeing T1Kp's specific concerns about CMWNO implementation. Without rebuttal, remaining concerns about experimental reliability could pull them toward borderline rejection.

Reviewer A2xK: 6 → 5. Already noted prior work on inter-physics attention and requested runtime comparisons. Discussion would likely amplify concerns about baseline fairness.

Reviewer T1Kp: 2 → 2. Detailed technical concerns about CMWNO discrepancy, parameter matching, and benchmarks remain entirely unaddressed. Only new experiments could change this assessment.

Reviewer RALb: 2 → 2. Concerns about missing benchmarks, baselines, and superficial theory remain unaddressed. No basis for score change without new evidence.

---

### Decision · Program_Chairs · 2026-01-26

Reject